# Vacancy driven surface disorder catalyzes anisotropic evaporation of ZnO (0001) polar surface

Zhen Wang [1,8], Jinho Byun [2,8], Subin Lee [1,3], Jinsol Seo[1,4], Bumsu Park[1,5], Jong Chan Kim [6], Hu Young Jeong [6], Junhyeok Bang [7] ✉, Jaekwang Lee [2] ✉ & Sang Ho Oh [1,4] ✉

The evaporation and crystal growth rates of ZnO are highly anisotropic and are fastest on the Zn-terminated ZnO (0001) polar surface. Herein, we study this behavior by direct atomic-scale observations and simulations of the dynamic processes of the ZnO (0001) polar surface during evaporation. The evaporation of the (0001) polar surface is accelerated dramatically at around 300 °C with the spontaneous formation of a few nanometer-thick quasi-liquid layer. This structurally disordered and chemically Zn-deficient quasi-liquid is derived from the formation and inward diffusion of Zn vacancies that stabilize the (0001) polar surface. The quasi-liquid controls the dissociative evaporation of ZnO with establishing steady state reactions with Zn and $O_2$ vapors and the underlying ZnO crystal; while the quasi-liquid catalyzes the disordering of ZnO lattice by injecting Zn vacancies, it facilitates the desorption of $O_2$ molecules. This study reveals that the polarity-driven surface disorder is the key structural feature driving the fast anisotropic evaporation and crystal growth of ZnO nanostructures along the [0001] direction.

Surface polar termination of ionic crystals consisting of alternate stacking of oppositely charged layers is unstable and exhibits high surface reactivity if the exposed surface charges remain uncompensated[1,2]. As such, a freshly cleaved polar surface is stabilized via various routes involving surface reconstruction and/or chemical modifications[3]. The hitherto known stabilization mechanisms include partial removal of surface atoms (vacancy formation)[4], hydroxylation[5], or the formation of step edges with opposite charge[6] or faceting toward a vicinal surface orientation[7].

The Zn-terminated (0001) surface of wurtzite ZnO consisting of alternate stacking of Zn and O layers is an archetypal example of polar surface. It has been shown that Zn vacancy ($V_{Zn}$) plays a central role in

stabilizing a pristine ZnO (0001) polar surface; a simple ionic model predicts that one $V_{Zn}$ in every four surface unit cells can stabilize the ZnO (0001) polar surface[2,8]. Various surface features have been reported for the ZnO (0001) polar surfaces, which are closely related to the formation of $V_{Zn}$[6,9]. For example, when annealed in vacuum, the ZnO (0001) polar surface tends to exhibit triangular or hexagonal pits of $V_{Zn}$ with exposing O-terminated step edges, which is favored by the Madelung energy[6,10]. It has been further shown that the regular array of these steps to a specific vicinal surface orientation, such as the $(10\bar{1}4)$, can compensate the dipole moment along the surface normal[7,11]. A recent study has well documented the stability of ZnO polar surfaces, which is critically governed by the entropic factors and associated

[1]Department of Energy Science, Sungkyunkwan University, Suwon 16419, Republic of Korea. [2]Department of Physics, Pusan National University, Busan 46241, Republic of Korea. [3]Institute of Applied Mechanics–Materials and Biomechanics, Karlsruhe Institute of Technology, Eggenstein-Leopoldshafen 76344, Germany. [4]Department of Energy Engineering, KENTECH Institute for Energy Materials and Devices, Korea Institute of Energy Technology (KENTECH), Naju 58330, Republic of Korea. [5]CEMES-CNRS, 29 rue J. Marvig, 31055 Toulouse, France. [6]UNIST Central Research Facilities, Ulsan National Institute of Science and Technology (UNIST), Ulsan 44919, Republic of Korea. [7]Department of Physics, Chungbuk National University, Cheongju 28644, Republic of Korea. [8]These authors contributed equally: Zhen Wang, Jinho Byun. ✉e-mail: jbang@cbnu.ac.kr; jaekwangl@pusan.ac.kr; shoh@kentech.ac.kr

structural disorder[12]. This implies that ZnO polar surfaces are non-stoichiometric and intrinsically rough at the atomic scale.

The ZnO (0001) polar surface evaporates abnormally fast at elevated temperatures[13]. Continuous removal of alternately layered Zn and O atoms from the (0001) surface during evaporation induces the instantaneous change of the surface structure and the charge compensation thereof. Therefore, one can expect that the evaporation influences or is influenced by the charge compensation mechanism. It has long remained a mystery that the evaporation of ZnO proceeds at a rate much higher than those predicted by thermodynamics[14,15]. Furthermore, the low temperature (<600 °C) evaporation of ZnO is highly anisotropic[13], where it evaporates predominantly by dissociation into Zn(g) and $O_2$(g), i.e., congruent evaporation via ZnO(s) = Zn(g) + $1/2 O_2$(g)[16,17]. Especially, the two polar surfaces of ZnO, positively charged Zn-terminated (0001) and negatively charged O-terminated (000$\bar{1}$), show greatly different evaporation behaviors; the (0001) surface shows about 10 times faster evaporation[13]. It is also well known that ZnO nanostructures grow abnormally fast along the [0001] direction[18–21]. We note that the surface energy of the (0001) polar surface is the largest among other low-index surface orientations of ZnO[22–24]. The surface with a higher surface energy usually has a higher growth rate, which will reduce its surface area, thereby decreasing the total surface energy of the whole system[25].

The origin of anomalous evaporation and growth behavior of the ZnO (0001) polar surface must be related with the intrinsic instability of surface[26,27]. One should consider the entropic contribution of surface atoms for the prediction of stable surface phase that gains thermodynamic stability during the evaporation or crystal growth of ZnO[28–30]. Especially, for the ZnO (0001) polar surface where a large number of $V_{Zn}$ and/or step edge sites exist to compensate the surface charges, the contribution of configurational entropy is unignorably large in determining the most stable surface structure at high temperatures[28]. As the configurational entropy lifts the energy difference and lowers the diffusion barriers between various surface phases at high temperatures, the surface may exhibit a highly dynamic or even disordered structure as a consequence of fluctuation between the phases[29].

It has also been reported that a non-equilibrium condition may exist in the ZnO (0001) polar surface where the evaporation is governed by the diffusion of defects or interface reactions[15,31]. Similarly, the formation of an impervious layer of non-stoichiometric oxide on the (0001) surface has been postulated, so that the evaporation reaction is essentially governed by the corresponding oxide[14]. The experimental observations, such as the change of color[14,32] and the dramatic enhancement of surface conductivity during evaporation[13], further support the evolution of nonequilibrium surface phase on the polar (0001) surface. In fact, the existence of amorphous phase has been speculated at the growth front of the polar (0001) surface during the atomic layer deposition of ZnO films[33,34]. While the existence of disordered and defective surface has been suspected, very little is known about the surface structure, the detailed mechanisms of evaporation[35] and crystal growth of ZnO[36] due to the difficulty of atomic-scale observation of high-temperature dynamic processes at the solid-vapor (SV) interface.

In this work, we unravel the mystery by directly observing the dynamic processes of ZnO evaporation at the atomic scale and rationalizing the observations by atomistic simulations. First, we directly show that the evaporation rate of the ZnO (0001) polar surface increases dramatically at around 300 °C with the spontaneous formation of self-catalytic quasi-liquid layer. The Zn-deficient quasi-liquid is derived from the formation and inward diffusion of $V_{Zn}$ that stabilizes a pristine ZnO (0001) polar surface. Appearing predominantly on the (0001) surface, the quasi-liquid controls the dissociative evaporation of ZnO with establishing steady-state reactions with Zn and

$O_2$ vapors in the atmosphere and the underlying ZnO crystal; while the quasi-liquid catalyzes the disordering of ZnO lattice by injecting $V_{Zn}$, it facilitates the desorption of $O_2$ molecules. Then, we show that the diffusion of $V_{Zn}$ triggers collective and coordinated dynamic motions of Zn and O atoms before the lattice instability sets in and drives the structural disordering of ZnO lattice. We found the gradual loss of the shear resistance of ZnO lattice with the growing concentration of $V_{Zn}$, which is analogous to the melting of a crystalline solid from the perspective of mechanical instability. Our direct observation combined with atomic simulations reveals that the polarity-driven surface disorder is the key structural feature responsible for the fast evaporation and growth along the $c$-axis of ZnO, especially the [0001] direction.

## Results

### Stabilization of ZnO (0001) surface by Zn vacancies

Transmission electron microscopy (TEM) samples were prepared along the [11$\bar{2}$0] zone axis of ZnO (0001) single crystal. Atomically thin, clean ZnO (0001) surface and other edge-on surface orientations, such as (10$\bar{1}$1), ($\bar{1}$011) and (10$\bar{1}$0) were obtained along the edge of TEM sample (Supplementary Fig. 1). The exposed (0001) polar surface is relatively narrow compared to other low index surface orientations due to its relatively high surface energy[22–24]. In high-resolution TEM (HRTEM) mode under a negative Cs imaging (NCSI) condition[37,38] Zn and O atomic columns are clearly resolved with small contrast delocalization at the surface (~40 pm) (Fig. 1a). Using the HRTEM image simulation we first determined the thickness of ZnO and then confirmed the linear relationship between the atomic column intensity and the number of Zn atoms at the NCSI imaging condition[39] (Supplementary Note 1 and Supplementary Fig. 2). Within the validity of the linear relationship between the atomic number and intensity, the occupancy of Zn atoms, equivalently the number of $V_{Zn}$, in the column has been assessed from the measured intensity of each column on the (0001) surface (Supplementary Fig. 2). In general, due to the unavoidable thickness gradient of TEM sample the number of Zn atoms measured near the surface is usually smaller than those in the bulk. However, the normalized intensity of Zn atoms decreased sharply in the surface termination layer, much more than expected from the thickness gradient, implying the presence of $V_{Zn}$ on the surface (Fig. 1c). Furthermore, the measured surface relaxation, in specific, the interlayer spacing, matches well with the theoretical calculation based on a $V_{Zn}$ model. All these experimental measurements of pristine ZnO (0001) surface in the TEM vacuum condition support the stabilization of the polar surface by $V_{Zn}$.

### Anisotropic dissociative evaporation of ZnO

The evaporation behaviors of various surface orientations of ZnO were investigated in-situ by HRTEM at temperatures from 150 to 700 °C to evaluate their orientation dependency. At temperature below 300 °C no measurable evaporation was detected from all surface orientations (Supplementary Fig. 1 and Supplementary Movie 1). The evaporation of ZnO became noticeable at 300 °C, especially from the (0001) surface (Fig. 2a, b, Supplementary Movie 2). The moving SV interface is often inclined towards the (10$\bar{1}$4) vicinal orientation with the formation of atomic steps, which is known to minimize the electrostatic energy of the polar ZnO (0001) surface[7,11] (Supplementary Fig. 3). Tracing the position of the SV interface of each surface orientation with time ($t$) shows that it moves linearly with time ($t$), indicating that the dissociative evaporation of ZnO proceeds at a constant rate (Fig. 2b). The constant rate implies that the evaporation is controlled by an interface reaction, i.e., the desorption of surface atoms. In this reaction-controlled regime, the evaporation rate of the (0001) surface is measured to be ~0.03 nm s$^{-1}$, which is apparently higher than other surface orientations (~0.01 nm s$^{-1}$). The evaporation of the (0001) surface becomes accelerated steeply by more than 10 times with the onset of quasi-liquid layer (corresponding to ~100 s in Fig. 2a, b), leading to

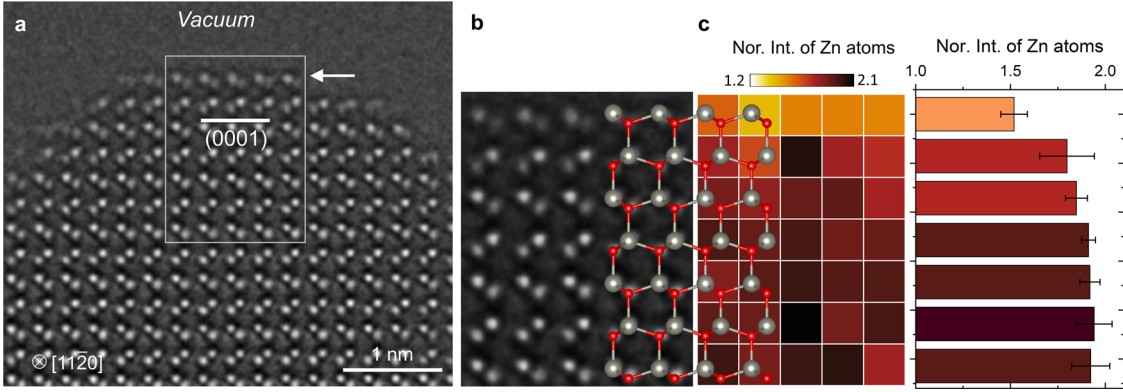

**Fig. 1 | HRTEM image of a pristine ZnO (0001) surface and quantitative analysis of the intensity of Zn atomic columns. a** HRTEM image of ZnO crystal recorded at a NCSI condition. The Zn-terminated (0001) surface in edge-on orientation is indicated by white arrow. **b** Magnified view of the region outlined by white lines in **a**. The region was selected for the quantitative analysis of the intensity of Zn atomic columns to assess the number of Zn atoms. **c**, Normalized intensity (Nor. Int.) of the Zn atoms displayed in 2D map for the individual Zn columns and histogram for the layer-averaged values with standard deviation. The atomic model is overlaid to guide the position of Zn (gray) and O (red) atoms.

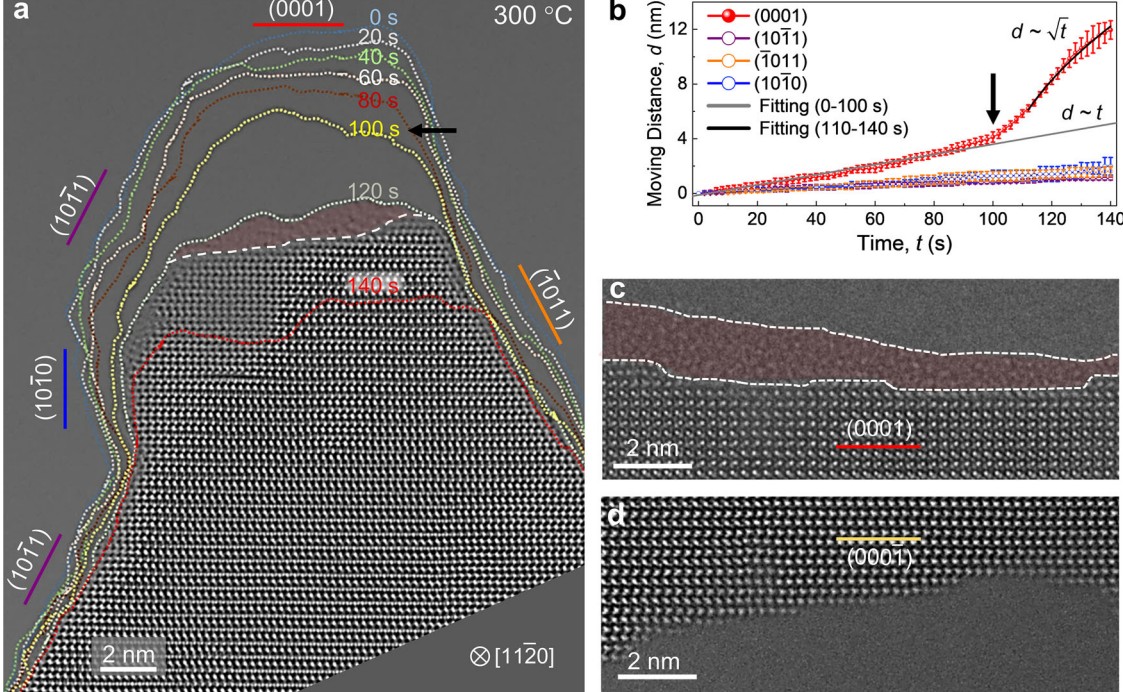

**Fig. 2 | Anisotropic evaporation of ZnO observed by in-situ HRTEM. a** Outlines of the polar (0001), semi-polar (10$\bar{1}$1), ($\bar{1}$011) and non-polar (10$\bar{1}$0) surface edges overlaid on HRTEM image tracing their movements during evaporation at 300 °C. The outlines of the edges traced in every 20 s for total 140 s are displayed in different colors. The quasi-liquid layer (highlighted in red) starts forming at ~100 s only on the (0001) surface. Referring to Supplementary Movie 2 for the corresponding real-time process. **b** Plot of the moving distance, $d$ of four major surface orientations during evaporation as a function of time, $t$. The black arrow marks the time when a quasi-liquid layer starts forming on the (0001) surface and the moving rate increases steeply afterwards. Before the formation of quasi-liquid layer the (0001) surface moves linearly with $t$. After the quasi-liquid layer formed, the (0001) surface moves with $t^{1/2}$ dependence, indicating that the evaporation is limited by a diffusion process. The error bars represent the standard deviation of data points along the direction in which they were averaged. **c**, **d**, HRTEM images of the (0001) and the (000$\bar{1}$) polar surfaces during evaporation, respectively. A quasi-liquid layer (highlighted in red) forms only on the (0001) surface but not on the (000$\bar{1}$) surface. All HRTEM images and movies are taken along the [11$\bar{2}$0] zone axis.

highly anisotropic evaporation of ZnO (Supplementary Fig. 4). Furthermore, the moving rate transits to exhibit a $t^{1/2}$ dependency, implying that the evaporation is controlled by a diffusion process. The HRTEM movie recorded at the NCSI condition (Supplementary Movie 3) confirms that the quasi-liquid layer is not an imaging artifact, which might be caused by contrast delocalization. We note that the quasi-liquid layer was formed only on the (0001) surface, not on any other surface orientations including the (000$\bar{1}$) polar surface

(Fig. 2c, d, Supplementary Fig. 5, and Supplementary Movie 4), implying that a surface disordering process takes place predominantly on the (0001) surface.

Upon nucleation on the (0001) surface, the quasi-liquid layer grows continuously and then reach a constant thickness at a given temperature. Intervening the SV interface between ZnO crystal and its vapor phases in the environment, the quasi-liquid layer mediates the dissociative evaporation; the (0001) surface is disordered first and

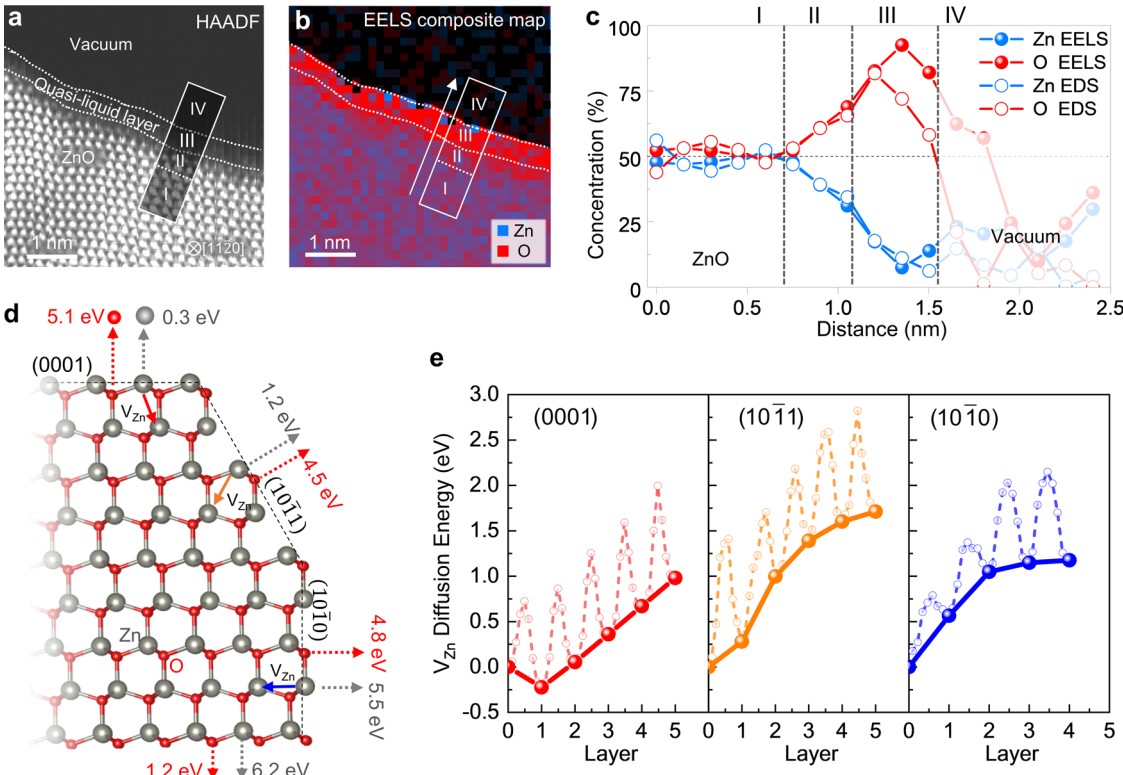

**Fig. 3 | Zn-deficient quasi-liquid layer on the ZnO (0001) polar surface. a** STEM-HAADF image showing the (0001) surface covered by quasi-liquid layer at 300 °C. **b**, Composite image constructed by selecting EELS Zn-L$_{2,3}$ and O-K edges. The region marked by I, II, III and IV indicates the bulk ZnO, subsurface Zn$_{1-x}$O, the quasi-liquid layer and vacuum, respectively. **c** Averaged composition profiles of Zn and O from the white lined box in **a** and **b** obtained by quantification of EELS data (solid symbols). The composition profiles obtained by quantification STEM EDS data by using Zn-K$_\alpha$ and O-K$_\alpha$ peaks are included (open symbols). For more details, refer to Supplementary Fig. 7 and Supplementary Note 2. Both EELS and EDS results consistently show that the quasi-liquid layer (region III) and subsurface (region II) are deficient of Zn, of which averaged Zn:O ratio is 0.2:0.8 and 0.4:0.6, respectively. The vacuum area with noise signal in **b** and **c** are shaded. **d** Atomic model depicting the calculated Zn and O desorption energy for the (0001), (10$\bar{1}$1), and (10$\bar{1}$0) surfaces. Grey and red circles represent Zn and O, respectively. The desorption energies are calculated by applying the density functional theory (DFT) at 0 K. **e** Energy variation of V$_{Zn}$ along the diffusion path from the ((0001), (10$\bar{1}$1), and (10$\bar{1}$0) surface towards the inner layers indicated as the solid arrow in **d**, which are calculated by the nudged elastic band method. Solid circles connected by solid line are the relative energy of the relaxed structure with V$_{Zn}$ on each layer; open circles connected by dash line are the energy variations in the transient structures.

then the resulting quasi-liquid evaporates instead of direct evaporation of ZnO crystal (Supplementary Movie 5). Once disordering and evaporation comes into a steady state at a given temperature, the quasi-liquid layer reaches an equilibrium thickness (e.g., 0.8 nm at 300 °C) and remains constant during the evaporation. The equilibrium thickness of the quasi-liquid layer increases linearly with temperature (Supplementary Fig. 6,a, b). The Arrhenius plot of the moving rate measured at various temperatures yields the activation energy of 0.34 ± 0.03 eV (Supplementary Fig. 6,c).

### Zn-deficient quasi-liquid layer on ZnO (0001) polar surface
The chemical composition of the quasi-liquid layer has been characterized by carrying out simultaneous electron energy loss spectroscopy (EELS) and energy dispersive x-ray spectroscopy (EDS) elemental mapping at 300 °C (Fig. 3a–c, Supplementary Fig. 7, Supplementary Note 2, Supplementary Table 1). Both EELS and EDS elemental maps and profiles revealed that the quasi-liquid layer (labeled as region III in Fig. 3a–c) is deficient of Zn compared to the ZnO bulk (region I). The EELS quantification yields that the average Zn:O ratio of the quasi-liquid layer is 0.2:0.8. Distinguished from the region I and III, the region II is defined as a Zn-deficient crystalline Zn$_{1-x}$O which extends around 0.4 nm from the quasi-liquid layer. The average Zn:O ratio of the region II is 0.4:0.6. The Zn and O profiles clearly show the existence of Zn deficiency that extends from the quasi-liquid layer to subsurface Zn$_{1-x}$O region (Fig. 3c), indicating the inward diffusion of V$_{Zn}$ from the (0001) surface. The Zn:O ratio averaged over the Zn

deficient surface region including the quasi-liquid layer and subsurface Zn$_{1-x}$O is 0.43:1, which is comparable to 0.5:1 of ZnO$_2$ peroxide. Note that ZnO$_2$ peroxide is the only stable solid phase in the Zn-O binary phase diagram other than ZnO[40]. However, ZnO$_2$ is known to be stable only up to 250 °C, above which it decomposes into ZnO and O$_2$ vapor phases[41].

### Anisotropic desorption energies and vacancy diffusion
The desorption energies of surface Zn and O atoms were calculated for various surface orientations of ZnO by applying density functional theory (DFT), which provide a clue to the anisotropic dissociative evaporation. The results show that the desorption energies of Zn is strongly orientation-dependent – 5.5 eV for the nonpolar (10$\bar{1}$0), 1.2 eV for the semipolar (10$\bar{1}$1), and 0.3 eV for the polar (0001) surface (Fig. 3d). Given that the local coordination of surface Zn atom is different for each surface orientation, the orientation dependency of the Zn desorption energy arises mainly from the electrostatic energy gain associated with the formation of negatively charged V$_{Zn}$ on the surface, which is the largest for the polar (0001) surface but decreases as the surface orientation deviates from the polar (0001), and the smallest for the nonpolar (10$\bar{1}$0) surface[10]. In contrast, the desorption energies of surface O atoms remain high for all orientations – 4.8 eV for the (10$\bar{1}$0), 4.5 eV for the (10$\bar{1}$1), and 5.1 eV for the (0001) surface. In addition, we also considered the O-terminated (000$\bar{1}$) polar surface and found that the calculated desorption energies for O and Zn atoms are 1.2 and 6.2 eV, respectively. The O desorption energy of the O-terminated

(000$\bar{1}$) polar surface is larger than the Zn desorption energy (0.3 eV) of the (0001) polar surface. These DFT results rationalize that only the Zn atoms can be preferentially desorbed from the (0001) polar surface.

Once a surface Zn atom is desorbed from the (0001) polar surface, the resulting $V_{Zn}$ experiences the driving force for diffusion toward the subsurface Zn layer, which further reduces the electrostatic energy. As shown in Fig. 3e, the energy is reduced by 0.22 eV when the $V_{Zn}$ resides in the first subsurface Zn layer. The diffusion barrier is calculated to 0.77 eV, which agrees well with the previous report by Erhart and Albe[42]. According to their DFT calculations, the out-of-plane diffusion of doubly negative charged $V_{Zn}$ to the first nearest neighbor Zn site is the most energetically favored migration path. The energy of $V_{Zn}$ in the second subsurface Zn layer becomes similar to that on the surface and increases further as the $V_{Zn}$ diffuses deeper into the bulk. In the other surface orientations, however, no such local minimum exists in the energy landscape along the diffusion path of $V_{Zn}$; the energy increases to 0.28 eV and 0.57 eV for the semipolar (10$\bar{1}$1) surface and the non-polar (10$\bar{1}$0), respectivley, when $V_{Zn}$ diffuses to the first subsurface Zn layer.

While the inward diffusion of $V_{Zn}$ is highly probable in the (0001) surface, we also considerd another competing but necessary process for the evaporation of ZnO, which is the desorption of the oxygen nearby $V_{Zn}$ forming a Schottky defect. As shown in Supplementary Fig. 8, the desorption energy of the oxygen (4.5 eV) is much higher than the energy barrier for $V_{Zn}$ inward diffusion (0.77 eV). The desorption energy of the O atoms remains large positive for all the other surface orientations as well. All these DFT calculations are put forwarded to rationalize the observed Zn deficiency growing inward the (0001) surface during the dissociative evaporation, that is, the preferential desorption of surface Zn atoms and consecutive inward diffusion of the resulting $V_{Zn}$.

The preferential desorption of Zn is not ascribed to electron beam-induced phenomena as the knock-on displacement and the radiolysis such as Knotek-Feibelman mechanism all predict the preferential desorption of O but not Zn[43,44] (Supplementary Note 3). Considering the calculated desorption energy of Zn (0.3 eV) for the (0001) surface, the thermal activation alone is expected to yield a measurable desorption rate at 300 °C (Supplementary Fig. 9,b). Moreover, as demonstrated by the control experiment and theoretical calculation, the desorption of Zn can occur on the (0001) surface under a reduced electron dose rate of ~$10^3$ e⁻ Å⁻² s⁻¹ (Supplementary Fig. 10) or purely by thermal activation at 600 °C without being assisted by electron beam (Supplementary Fig. 9,a). Once the surface Zn atoms are desorbed, then the inward diffusion of $V_{Zn}$ becomes the rate-limiting step, which is in accord with the measured $t^{1/2}$ dependency of the moving rate (Fig. 2b). The inward diffusion of $V_{Zn}$ can be easily activated at 300 °C due to the low activation energy (0.77 eV), which is compared well with the measured activation energy of ~0.34 eV from the Arrhenius plot of evaporation rate (Supplementary Fig. 6,c). The measured activation energy smaller than the calculated diffusion barrier is attributable to the catalytic effects of quasi-liquid layer and/or the electron beam stimulation (Supplementary Note 3). In contrast, since $V_{Zn}$ hardly diffuses into the bulk ZnO from the other non-polar (10$\bar{1}$0) and semi-polar (10$\bar{1}$1) surfaces, it is predicted to exhibit the desorption of surface Zn and O atoms one by one at a relatively high temperature without the accumulation of $V_{Zn}$ in the bulk region.

## $V_{Zn}$ diffusion induced lattice instability

The Zn-deficient subsurface region of the polar (0001) surface exhibits abnormally large displacement of Zn and O columns before the loss of long-range order (Supplementary Movie 6). The mean displacement of Zn atomic columns measured by tracing the peak position of atomic column intensity was ~11.06 ± 3.39 pm (Supplementary Fig. 11), which far exceeds the value measured from Zn atomic columns in the bulk region (4.85 ± 0.9 pm) at 300 °C (Supplementary Fig. 12). Furthermore,

the displacements of the Zn and O columns are collectively coordinated with neighboring column; for a given displacement of Zn or O column, those of adjacent columns tend to point similar directions, resulting in a laminar flow of displacement vectors (Fig. 4a, b, Supplementary Figs. 12,b, 13, Supplementary Movie 7). It is interesting to note that the displacement vectors, although fluctuate constantly with time, point more frequently the in-plane directions; the measurement of displacement vector components shows that the in-plane component is about 3 times greater than that of the out-of-plane component (Fig. 4c). Although the displacements of the Zn and O atomic columns induce local lattice distortion of the corresponding unit cell, they do not induce global external lattice strain (Supplementary Fig. 14, Supplementary Movie 8).

The dynamic atomic displacements in the Zn-deficient subsurface region are closely related to the diffusion of $V_{Zn}$. To assess the occupation and distribution of $V_{Zn}$ in each Zn column, the intensity of Zn atomic columns has been measured quantitatively and traced their changes with time on a series of NCSI HRTEM images. As provided as color-coded $V_{Zn}$ maps in Fig. 4b (Supplementary Fig. 13) and the density profile in Fig. 4d, the $V_{Zn}$ concentration is high near the surface and decreases gradually towards the ZnO bulk. For example, the near-surface Zn columns contain three $V_{Zn}$ (purple pixels) and those away from this region contain two $V_{Zn}$ (red pixels) or one $V_{Zn}$ (orange pixels) (Fig. 4b). Direct comparison of the $V_{Zn}$ map with the displacement map yields a high degree of correlation, i.e., the Zn columns containing a high density of $V_{Zn}$ exhibit a large displacement (Supplementary Fig. 13, Supplementary Movie 7).

Given that the displacement and disordering of ZnO lattice is driven by diffusion and accumulation of $V_{Zn}$, the lattice instability of ZnO caused by $V_{Zn}$ has been studied by calculating the elastic constants of Zn-deficient $Zn_{1-x}O$ by applying DFT (Supplementary Note 4, Supplementary Figs. 15-16). Wurtzite ZnO with a hexagonal crystal system has five independent elastic constants ($C_{ij}$, namely $C_{11}$, $C_{12}$, $C_{13}$, $C_{33}$, $C_{44}$) due to the additional relation $C_{66} = (C_{11} - C_{12})/2$. While the elastic constants $C_{11}$ and $C_{33}$ represent stiffness against principal strains, $C_{66}$ and $C_{44}$ reflect the resistance against shear along the in-plane direction on the (0001) plane and the (10$\bar{1}$0)plane, respectively. For a stable hexagonal structure, its five independent elastic constants should satisfy the well-known Born stability criteria, i.e., $C_{11} - |C_{12}|>0$, $(C_{11}+C_{12})C_{33} - 2C_{13}^2 >0$ and $C_{44} > 0$[45]. The calculated elastic constants with respect to the Zn deficiency $x$ in $Zn_{1-x}O$ shows that both shear elastic constants $C_{66}$ and $C_{44}$ decrease almost linearly with the Zn deficiency (Fig. 4e). Interestingly, $C_{66}$ associated with the in-plane shear stiffness of the (0001) plane decreases with $V_{Zn}$ more rapidly than $C_{44}$, governing the lattice stability of $Zn_{1-x}O$. This shear instability predicts that $Zn_{1-x}O$ softens more easily along the in-plane directions of (0001) plane with growing concentration of $V_{Zn}$, which is compared well with the experimentally observed in-plane dominant displacement field of Zn atomic columns. From the viewpoint of $V_{Zn}$ diffusion, the inward diffusion of $V_{Zn}$ from the surface via the energetically favored out-of-plane migration path is also expected to cause the in-plane dominant displacement field as it produces the lattice distortion predominantly along the in-plane directions.

In the DFT calculations, the $Zn_{1-x}O$ structure with $x > 0.2$ undergoes large distortion, so that we could not determine the Zn deficiency $x$ satisfying the condition $C_{66}$ or $C_{44} = 0$, i.e., the criterion for the complete loss of long-range order or. equivalently, melting. Based on the almost linear dependence of $C_{66}$ and $C_{44}$ with $V_{Zn}$, the extrapolation of the fitting curve to the condition $C_{66} = 0$ and $C_{44} = 0$ yields $x = 0.25$ and 0.42, respectively (Fig. 4e). We note that the elastic constants were calculated in the zero temperature limit. Because the elastic constants tend to decrease with temperature[46,47], the lattice disorder can initiate in the Zn deficiency $x < 0.25$ at 300 °C. This result goes in-line with the increase of the thickness of quasi-liquid layer with temperature.

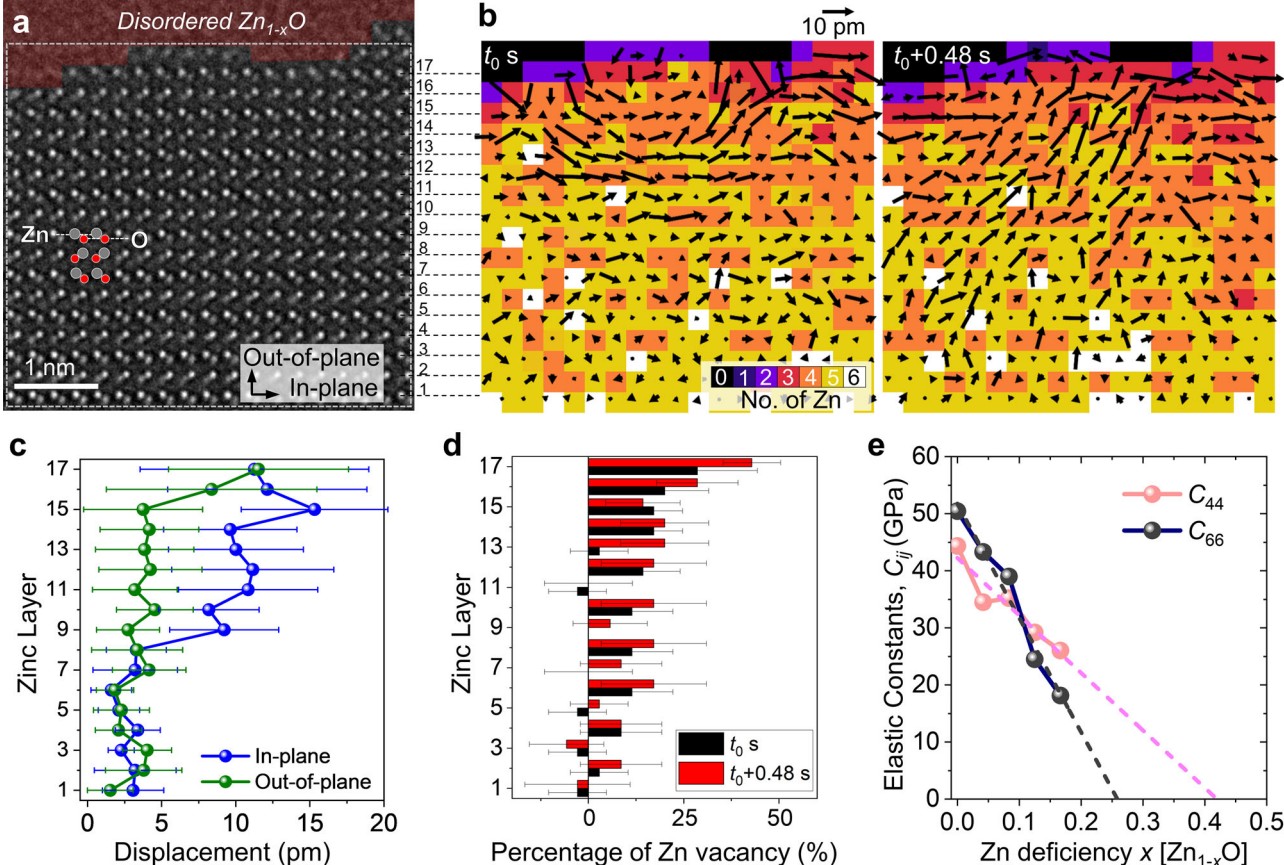

**Fig. 4 | $V_{Zn}$ diffusion induced lattice dynamics and instability of subsurface $Zn_{1-x}O$ during evaporation. a** In-situ NCSI-HRTEM snapshot image of ZnO during anisotropic dissociative evaporation from the ZnO (0001) polar surface at 300 °C. The quasi-liquid layer formed on the (0001) surface is highlighted in red. Bright and dim atomic columns correspond to Zn and O columns, respectively. The Zn layers are numbered from the bulk to the surface. **b** Time series displacement vector (black arrow) and atom count maps (colored pixels) of Zn columns. The initial time of measurement is defined as $t_0$. The number of Zn atoms in each column was determined by quantitative analysis of the Zn column intensities (Supplementary Fig. 11). The majority of Zn columns (yellow pixels) contains five Zn atoms. The reference positions for displacement map were the averaged positions over selected time spans. The magnitude of vector is multiplied by 30 times for better visualization. A high degree of correlation between $V_{Zn}$ and displacement of Zn column is found; the Zn columns containing less Zn atoms, equivalently more $V_{Zn}$, exhibit large displacement in the subsurface region (Supplementary Movie 7). **c** Averaged in-plane and out-of-plane components of the displacement vector for each Zn layer from the bulk towards the surface. The in-plane displacement component is around 3 times larger than the out-of-plane component in the subsurface region. The displacement of Zn columns was averaged over 5 s. The error bar is the standard deviation of displacement for each Zn layer in the in-plane direction. **d** Averaged percentage of $V_{Zn}$ for each Zn layer from the bulk towards the surface. The two time-series Zn atom count maps shown in **b** were used for the plot. The error bar is the standard deviation of $V_{Zn}$ percentage for each Zn layer in the in-plane direction. A higher concentration of $V_{Zn}$ in the subsurface region is clearly visualized and correlated with the pronounced in-plane displacement of Zn. **e** Calculated elastic constant $C_{44}$ and $C_{66}$ as a function of Zn deficiency ($x$) in $Zn_{1-x}O$. $C_{66}$ associated with the in-plane shear stiffness of the (0001) plane decreases with $V_{Zn}$ more rapidly than $C_{44}$. The extrapolation based on a linear fit predicts $C_{66} = 0$ when $x = 0.25$.

## Lattice disordering and dissociative evaporation of $Zn_{1-x}O$

We traced the intensity and displacement of Zn atomic columns in a series of HRTEM images until the long-range order breaks down. Figure 5a shows the time-sequence HRTEM images where the interface between the quasi-liquid layer and crystalline $Zn_{1-x}O$ is marked and two regions representing the $V_{Zn}$-induced disordering (red box) and ordinary thermal vibration (blue box) are defined. At the initial time of measurement (defined as $t_0$), both regions were away from the interface. As the time goes, the interface is moving down and gets close to the red box (15 s later), and later the region gets disordered completely (26 s later). On the contrary, the atomic columns in the blue box remain in the crystalline phase. The trajectory of Zn atomic columns traced for the time span of 26 s visualizes relatively larger displacements in the red box compared with those in the blue box (Fig. 5b). Using the measured trajectory, we calculated the mean square displacement (MSD) of the atomic columns in the red box and plotted their change with time (Fig. 5c). The time dependency of the measured MSD is not simply accounted for by the behavior of a single

state as the displacement of $Zn_{1-x}O$ increases with the growing density of $V_{Zn}$ and, furthermore, it undergoes disordering analogous to glass transition or melting. Nonetheless, one can clearly notice that the MSD is not bound by a plateau but increases linearly with time up to $t = 15$ s and then abruptly as the interface was approached. This behavior is in accord with what is expected from the Lindemann criterion of melting based on the vibration instability hypothesis[48], which predicts that the melting occurs when the mean square amplitude of vibration reaches a fraction (~10%) of the nearest-neighbor spacing (1.96 Å for ZnO). The MSD of Zn atomic columns measured just before the loss of long-range order reaches ~8% of the nearest-neighbor spacing (Fig. 5c). Tracing the intensity variation of the atomic columns with time reveals that the intensity of Zn columns decreases rapidly from 15 s to the background level, indicating the loss of Zn atoms in the column due to the increase of $V_{Zn}$, resulting in the collapse of long-range order. Our real-time atomic-scale imaging and analysis combined with DFT calculation confirmed that the $V_{Zn}$-driven disordering of the polar ZnO (0001) surface

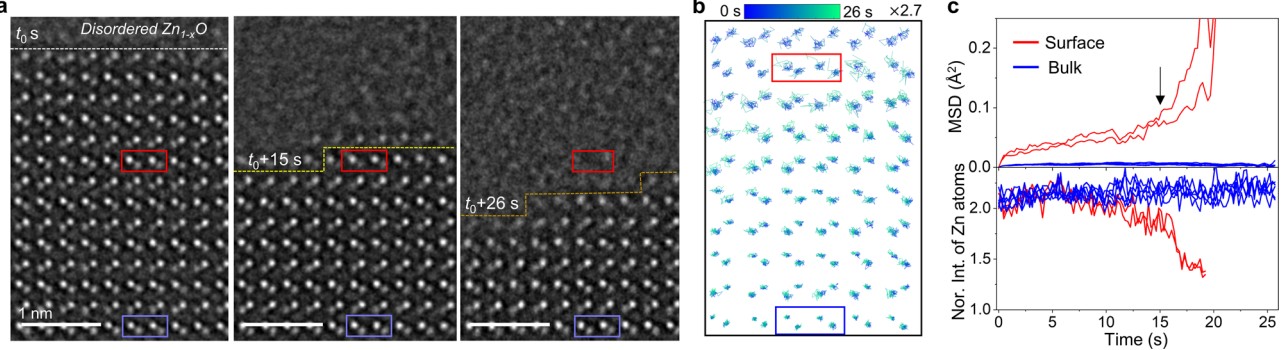

**Fig. 5 | Atomic-scale disordering process during evaporation. a** Time-series NCSI-HRTEM snapshot images of the (0001) surface during evaporation at 300 °C (Supplementary Movie 5). The interface between the quasi-liquid layer and crystalline $Zn_{1-x}O$ is marked and two regions representing the $V_{Zn}$-induced disordering (red box) and ordinary thermal vibration (blue box) are defined. **b** Trajectory of Zn and O atomic columns traced for the time span of 26 s. The trajectory visualizes relatively larger displacements of Zn and O columns in crystalline $Zn_{1-x}O$ (red box) compared with those in the blue box. **c**, MSD measured for the Zn atomic columns

in red and blue box plotted with time (top). MSD increases linearly with time up to $t$ = 15 s and increases abruptly as the interface was approached. The MSD of Zn atomic columns is measured to be 15 pm just before the loss of long range, which is about 8% of the nearest-neighbor spacing. Intensity of the Zn columns in red and blue box plotted with time (bottom). The intensity of Zn columns in $Zn_{1-x}O$ (red box) decreases rapidly from 18 s to the background level, indicating the loss of Zn atoms in the column due to the increase of $V_{Zn}$.

during evaporation is described well by two melting criteria, i.e., the vibration instability according to the Lindemann criterion and the shear instability according to the Born criterion.

To validate the role of $V_{Zn}$ in disordering and dissociative evaporation of the ZnO polar (0001) surface, Born–Oppenheimer molecular dynamics (MD) simulations have been carried out in the microcanonical ensemble (*NVE*), where the number of particles (*N*), the volume (*V*), and the energy (*E*) are conserved (Fig. 6a, Supplementary Fig. 17-18, Supplementary Movie 9). The simulation cell was prepared by introducing $V_{Zn}$ to the first three Zn layers to make the composition the same as that of the quasi-liquid. The MD simulations show that $V_{Zn}$ induces the lattice distortions and disordering, resulting in the formation of disordered layer and the desorption of oxygen molecules ($O_2$) from the surface (Fig. 6a). The disordered layer is clearly distinguished from the bulk by radial distribution function (RDF). The RDF of the disordered layer exhibits a single main peak at about 2 Å followed attenuated peaks, which is a characteristic short-range order of the liquid state (Fig. 6b). We also confirmed that the disordered $Zn_{1-x}O$ can be distinguished from the bulk ZnO by EELS O-K edge fine structure analysis as well (Fig. 6c). The fine structure of EELS O-K edge obtained from the quasi-liquid layer revealed an additional peak at a higher energy (544 eV, black arrow in Fig. 6c), which does not exist in wurtzite ZnO crystal. The calculated unoccupied oxygen density of states (DOS) from quasi-liquid $Zn_{1-x}O$ layer shows a similar peak appearing at the same energy. Further detailed investigation of the unoccupied oxygen DOS with different atomic configurations reveals that the high energy peak arises from the lack of long-range order, especially due to the absence of the 3rd nearest neighbor (N.N.) (Zn) in RDF induced by $V_{Zn}$ (Supplementary Fig. 18).

The layer-by-layer DOS shows that the quasi-liquid is metallic, so that in principle its free electrons can screen the surface charges of the (0001) polar surface (Fig. 6d, Supplementary Fig. 19). However, the metallicity of quasi-liquid layer should not be regarded as the ultimate mechanism for the polarity compensation. As shown in Fig. 1, a pristine ZnO (0001) polar surface can be stabilized readily by forming $V_{Zn}$ and/ or atomic steps at temperatures below 300 °C without lattice disordering or quasi-liquid layer formation. Rather, the formation of quasi-liquid layer should be considered as a result of the growing Zn deficiency with temperature due to the preferential desorption of Zn and vacancy diffusion which are thermally activated processes driven by the polarity compensation. Considering the surface state of ZnO (0001) at the condition where quasi-liquid layer evolves, the surface polar termination cannot be defined as an ideal Zn atomic layer

since the surface roughness is constantly evolving during the dissociative evaporation. Furthermore, the $V_{Zn}$ diffuses away from the surface toward the bulk, resulting in an extended distribution of $V_{Zn}$. As a consequence, the surface relaxation (the interlayer spacing, in specific) and the ionic polarization of the ZnO (0001) covered by quasi-liquid layer are modified noticeably compared to the pristine surface; the interlayer relaxation, once strongly confined to the first layer of pristine (0001) surface, is extended over multiple layers from the surface due to the extended distribution of $V_{Zn}$ and the ionic polarization decreases and disappears near the surface.

The Zn-deficient quasi-liquid layer catalyzes the dissociative evaporation of ZnO by greatly reducing the desorption energy of oxygen. While it is difficult to reproduce the experimentally observed disordered $Zn_{1-x}O$ phase in the theoretical calculation, the desorption of oxygen calculated from crystalline $ZnO_2$ phases with a similar composition helps us to estimate the desorption energy of oxygen (Supplementary Note 5 and Supplementary Fig. 20). Compared to the high desorption energy of oxygen (5.1 eV) in the (0001) surface, we found that the desorption energy of oxygen is reduced dramatically below 1 eV in the crystalline $ZnO_2$ phases, facilitating the dissociative evaporation of $Zn_{1-x}O$ in Zn and $O_2$ molecules (Fig. 6e).

## Discussion

Our direct atomic-scale observation revealed that the surface disorder driven by the formation and diffusion of $V_{Zn}$ catalyzes the dissociative evaporation of ZnO (0001) polar surface. Appearing predominantly on the (0001) polar surface at around 300 °C, the quasi-liquid layer controls the dissociative evaporation of ZnO with establishing steady-state reactions with Zn and $O_2$ vapors in the atmosphere and the underlying ZnO crystal; while the quasi-liquid layer catalyzes the disordering of ZnO lattice by injecting $V_{Zn}$, it facilitates the desorption of $O_2$ molecules. With growing concentration of $V_{Zn}$, ZnO undergoes structural disordering which is analogous to the first order melting transition from the perspective of mechanical as well as vibrational instability. While the direct observation of the evaporation of ZnO polar surface is important to understand the surface reconstruction process, it can also provide valuable insights into the anisotropic growth mechanism of ZnO nanostructures since the growth is a reverse process of the evaporation. Our work emphasizes the important role of cation vacancy in stabilizing a polar oxide surface and its significant contribution to entropy, especially, at a high temperature during dynamic processes such as structural reconstruction or phase transformation[12,29]. The entropic factors also play a key role on the

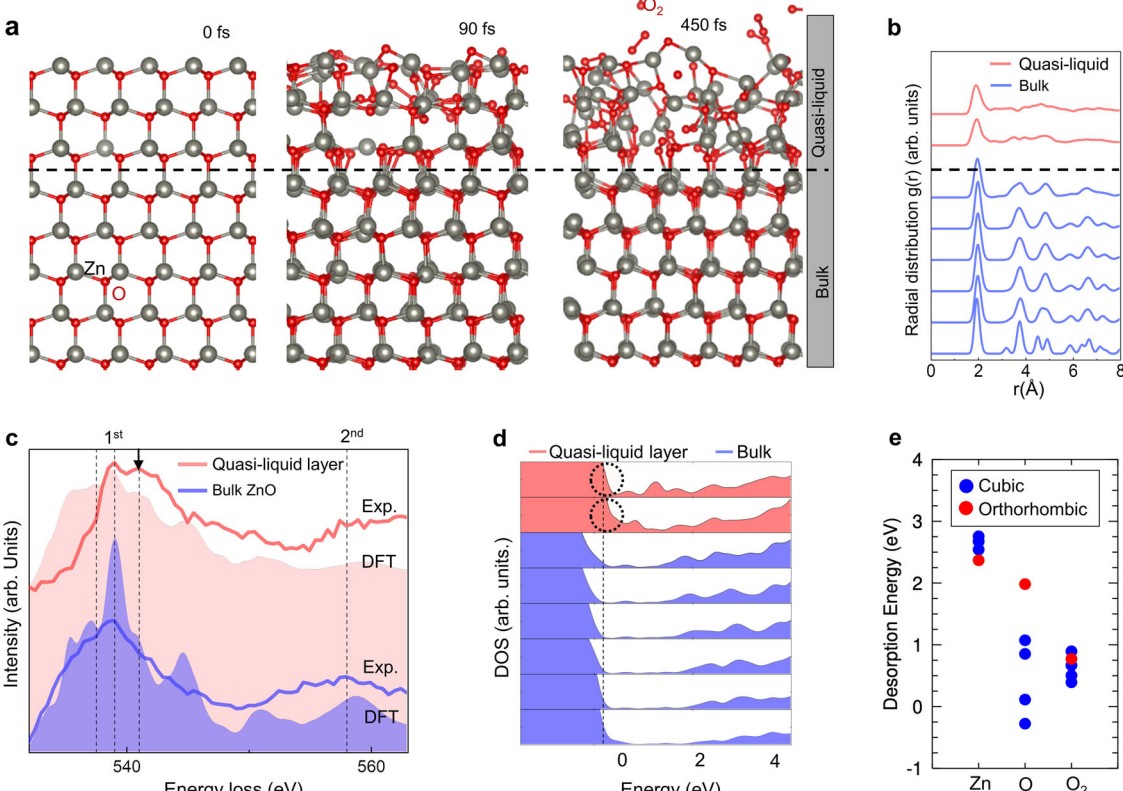

**Fig. 6 | Characteristics and role of quasi-liquid layer in dissociative evaporation of ZnO (0001). a** Ab initio MD simulation snapshots showing the disordering of Zn-deficient $Zn_{1-x}O$ (top three bilayers above the dash line). $V_{Zn}$ were introduced to the top three Zn layers to make the composition the same as that of the quasi-liquid layer. Oxygen molecules ($O_2$) desorbed from the surface are indicated. Detailed results of MD simulation are given in Supplementary Figs. 17, 18, Supplementary Movie 9. **b** RDF of each Zn and O bilayer of the simulated structure in **a**. The disordered quasi-liquid layer surface (red) shows a characteristic short range order of liquid state. The dash line differentiates the quasi-liquid ($V_{Zn}$) from the bulk region. **c**, Experimental (Exp.) EEL spectra of O-K edge (blue and red line) and

calculated unoccupied oxygen density of states (DOS) from DFT (blue and red area). The quasi-liquid layer in both experimental EELS O-K and calculated unoccupied oxygen DOS shows additional peak at around 544 eV (black arrow), which does not exist in bulk ZnO. **d** Layer-by-layer DOS of each Zn and O bilayers. The quasi-liquid $Zn_{1-x}O$ layers (red) show a metallic character (dashed circles) whereas bulk ZnO layers (blue) show an insulating character. **e** Calculated desorption energy of Zn, O and $O_2$ in ZnO and $ZnO_2$ by DFT. The cubic and orthorhombic $ZnO_2$ are represented as blue and red, respectively. Desorption energy of O in form of $O_2$ molecule is reduced dramatically below 1 eV in the crystalline $ZnO_2$ phases (Supplementary Figs. 20).

O-terminated $(000\bar{1})$ polar surface. Similar to the (0001), the $(000\bar{1})$ polar surface is stabilized via non-stoichiometric surface reconstruction involving the formation of oxygen vacancies. However, different from the (0001), the surface disorder at high temperature does not result in the formation of quasi-liquid layer on $(000\bar{1})$ polar surface (Supplementary Fig. 21). This is due to the different desorption behavior of Zn atoms from the $(000\bar{1})$ surface and the lack of an oxygen deficient solid phase in the Zn-O binary system.

## Methods

### TEM sample preparation
TEM samples were prepared using a ZnO (0001) single crystal (CrysTec GmbH, Germany) along the $[11\bar{2}0]$ zone axis by using focused ion beam (FIB, Helios NanoLab 450, FEI) lift-out technique. As-FIB prepared TEM samples were further cleaned by $Ar^+$ ion milling and plasma cleaning to reduce the surface damages and contaminations. In-situ TEM heating experiments were performed using a double-tilt MEMS chip-based heating holder (DENSsolutions).

### In-situ HRTEM imaging
A double Cs-corrected field-emission TEM (JEOL, JEM-ARM300 CF) operated at 300 kV was used for in-situ heating experiments. The vacuum of TEM column was around $2 \times 10^{-5}$ Pa. An NCSI technique was applied to resolve Zn and O columns during the dissociative evaporation of ZnO[37,38]. The objective lens aberrations of the microscope

were measured using the Zemlin tableau method from thin amorphous area and were tuned shortly before the image acquisition[49]. Spherical aberration was tuned iteratively until it converged to around −13 μm, which is the optimal condition for NCSI imaging[38]. According to the measurement, the images were recorded using a defocus value (C1) + 4–6 nm with the two-fold astigmatism (A1) < 1 nm, three-fold astigmatism (A2) < 25 nm, and axis coma (B2) < 10 nm. The electron dose rate was controlled to be around $2 \times 10^4$ $e^-$ $Å^{-2}$ $s^{-1}$ to give proper signal-to-noise ratio (SNR) and to reduce e-beam damages. At the NCSI HRTEM imaging condition high image contrast of (bright) atomic columns could be obtained on (dark) background with a minimum contrast delocalization (~40 pm), allowing the highly precise determination of the atomic column position within picometer precision (~4 pm). Real-time movies were acquired by using a 4096 × 4096 pixels CMOS camera (OneView Camera, Gatan) at 25 frames per second (fps) and also at 100 fps with binning to capture the fast evaporation process.

### In-situ STEM EELS and EDS elemental mapping
STEM EELS $Zn-L_{2,3}$ and O-K edges were obtained at 300 kV using an EEL spectrometer (Gatan GIF Quantum ER 965, USA) with an energy resolution of 0.7 eV. The probe convergence angle and collection angle were 23 and 36 mrad, respectively. The Zn and O concentration were determined by the quantification of EELS $Zn-L_{2,3}$ and O-K edges. STEM EDS data were acquired simultaneously with EELS data by using an EDS

detector with the area of 100 mm$^2$ (JEOL). The specimen drift was corrected during the EDS and EELS acquisition. The Zn and O elemental maps were constructed by integrating the signal from Zn-K$_\alpha$ and O-K$_\alpha$ characteristic X-rays peaks. The quantitative analysis of the obtained EDS and EELS data was processed by Gatan DigitalMicrograph software.

## HRTEM image analyses

For highly precise tracing of atomic column positions, rigid registration was applied rigorously to eliminate the image drift in in-situ HRTEM movies. The inner bulk area with no significant change of contrast was selected as reference area for registration. Iterative registration was done for several time to minimize measured drift less than 10$^{-5}$ nm, which ensures the high precision position tracking. Several software was used for post image processing of the time sequence HRTEM images. The atom column centers and intensities were determined precisely from the bright contrast which separately extracts the position of Zn and O atom columns. To track the dynamic motions of atoms during the evaporation followed by disordering, MATLAB codes with two-dimensional Gaussian fitting algorithm were used to determine Zn atom position with subpixel precision frame by frame[50]. As the precision of fitting is determined by the SNR of images, 6 frames were summed to enhance the SNR while maintaining a relatively high temporal resolution (0.24 s). No filter was applied to avoid any artifact. Zn columns with higher contrast gave better precision than O columns. The standard deviation (S.D.) of each atom position from the averaged position is used to evaluate the thermal vibration amplitude, which is defined as

$$SD = \sqrt{\frac{1}{N}\sum_1^N \delta_i^2} \qquad (1)$$

$$\delta_i^2 = (x_i - \bar{x})^2 + (y_i - \bar{y})^2 \qquad (2)$$

where $i$ is the frame number, $N$ is the total number of frames, $x_i$ and $y_i$ represent the coordinates of atom column position in the $i$-th frame, $\bar{x}$ and $\bar{y}$ represent the coordinates of atom column position averaged over $N$ frames, and $\delta_i^2$ is the squared deviation of atom column position in the $i$-th frame from the averaged position. To make the trajectories of atom motions the extracted time series of each column position were linked together using self-programed scripts. Then, the MSD for each column was calculated based on its single trajectory[51]. The tracked coordinates of Zn and O atoms from time-series frames were exported to OVITO software for calculation and visualization of displacement map[52]. The reference positions for displacement map were the averaged positions over selected time spans.

## HRTEM image simulation

HRTEM image simulations were carried out using the open-source program QSTEM. The following simulation parameters determined from the experiments were applied: spherical aberration coefficient $C_s = -13$ μm, focal spread $\Delta f_s = 0.5$ nm, convergence angle $\alpha = 0.6$ mrad. Quantitative analysis was done by an interactive iterative digital image matching (IDIM) process from intuitive graphical user interface (GUI) HREM-DIMA software. Local optimized imaging parameters like defocus, the thickness of the concerned area, specimen tilt from the exact [11$\bar{2}$0] zone axis were determined by maximizing the cross-correlation coefficient between simulated image and experimental images. To ensure precise matching of the contrast between experimental and simulated images, the camera modulation transfer function (MTF) has been taken into account for the image simulation[53], which is regarded as major factor to solve the Stobbs-factor problem[54]. The MTF profile of the OneView (Gatan) camera was calculated by the sharp edge images from aperture[55].

## DFT calculations

The optimized geometries and relevant total energies were obtained via spin-polarized DFT calculations, as implemented in the Vienna ab initio simulation package[56]. We used the projected augmented wave potentials[57] and the local density approximation[58] with on-site Coulomb energy (LDA + U)[59]. Here U = 4.7 eV is set at Zn $d$ orbitals. This LDA + U method has been employed in previous works, and the results have shown that the method describes defects in ZnO fairly well[60]. We also checked the validity of the LDA + U method by comparing with the Heyd-Scuseria-Ernzerhof (HSE) functional calculation and found that the two results are in good agreement (see Supplementary Note 6, Supplementary Table 2, Supplementary Fig. 19). The wave functions were expanded in a plane wave basis set up to a cut-off energy of 400 eV. Atomic positions were fully relaxed until the residual forces were less than 0.01 eV/Å.

To investigate atomic desorption and vacancy diffusion, we employed periodic slab models containing 10 atomic layers for the (0001) and (10$\bar{1}$1) surfaces and 8 atomic layers for (10$\bar{1}$0) surface, as shown in Supplementary Fig. 8. The slabs were separated by the vacuum gap of 25 Å. The 4×2×1 Monkhorst-Pack k-point mesh was used for Brillouin-zone integration. If the charge transfer occurs between two opposite surfaces, the strong and constant electric field is formed inside ZnO. To prevent the charge transfer, we passivate the O and Zn dangling bonds on the one surface using 0.5 and 1.5 fractionally charged pseudo-hydrogen atoms, respectively. The method was widely used in previous works[61,62]. Similar approaches have also been adopted for ZnO[9,63]. We also checked the local potential in our slab model and found that the dipole field is fully compensated. For each surface, the desorption energy of the $X$ atom $E_{DE}(X)$ is calculated by $E_{DE}(X) = E(S : V_X) + E_{ref}(X) - E(S)$, where $E(S : V_X)$ and $E(S)$ are the total energies of the surface with $V_X$ and the pristine surface, respectively, and $E_{ref}(X)$ is the reference energy of the $X$ atom, i.e., $E_{ref}(Zn)$ and $E_{ref}(O)$ are the single atom energies of the Zn metal and O$_2$ molecule, respectively.

To model the Zn-deficient quasi-liquid layer on ZnO (0001) surface, we placed the ZnO$_2$, of which Zn/O composition is similar to that of the observed quasi-liquid layer, on the (0001) surface slab model. We considered the two stable ZnO$_2$ crystals, i.e., cubic and orthorhombic, as shown in the Supplementary Fig. 20. The thickness of the two ZnO$_2$ layers are 7.2 Å for cubic and 9.1 Å for orthorhombic.

The elastic constants were calculated using a supercell containing 96 host atoms of the wurtzite ZnO, and the 2×2×2 Monkhorst-Pack k-point mesh was used for Brillouin-zone integration. Note that it is very difficult to calculate the elastic constants directly using the anisotropic slab cells with the gradient of the V$_{Zn}$ concentration. Even in the same V$_{Zn}$ concentration, the elastic constants are slightly changed depending on the V$_{Zn}$ distribution in the supercell. For better statistics, we considered the three different supercells with randomly distributed V$_{Zn}$ for each concentration, as one example is illustrated in Supplementary Fig. 15, and the calculated elastic constants for the samples are averaged.

## Ab initio MD simulations of ZnO slab with quasi-liquid layer

We have modeled a Zn deficient wurtzite ZnO slab, oriented in the (0001) direction, consisting of 261 Zn and 300 O atoms. 39 Zn atoms were removed from the top 3 Zn-O bilayers to reproduce the Zn deficiency observed experimentally by EDS and EELS. The size of slab in the x, y, and z directions was 16.02 Å × 16.67 Å × 40.67 Å with a vacuum region of 16 Å, which ensures that the interaction between periodic images can be safely avoided. The bottom layer of the slab is frozen at bulk configuration.

Ab initio MD simulations were performed by using the CP2K package[64] where nuclear motion is described classically and the electronic structure is described quantum mechanically by DFT. The ab initio MD Born-Oppenheimer trajectories have been propagated using QUICKSTEP[65] electronic structure module as implemented in CP2K.

Gaussian and plane-wave method of QUICKSTEP module was used to study the electronic structure during the geometry optimization.

LDA+U method is well established for crystalline ZnO. However, unlike the crystalline ZnO, for metallic quasi-liquid ZnO empirical parameter U for LDA+U is not known. For that reason, the standard Perdew-Burke-Ernzerhof (PBE) functional was used together with Goedecker-Teter-Hutter (GTH) pseudopotentials[66] for ab initio MD simulations of a ZnO slab with quasi-liquid layer. The Kohn-Sham orbitals were expanded in Gaussian functions with molecularly optimized double-zeta polarized basis sets (MOLOPT-DZVP)[67], whereas the electron density was expanded in plane-wave with a cutoff of 380 eV[68]. The Brillouin zone was sampled at the Gamma (Γ) point only. Electronic relaxations were performed with a convergence threshold of $2.7 \times 10^{-5}$ eV per electronic step.

The system relaxes with the time step of 3 fs during 1.6 ps with a microscopic (NVE) ensemble. The initial velocities were drawn from the Maxwell-Boltzmann distribution at 600 K, and the temperature was stabilized at ~1000 K after 0.15 ps. To identify the changes in crystal structure, the RDFs were carried out from the structure obtained at 0.45 ps of MD simulation.

### Layer-by-layer DOS calculation of ZnO slab with quasi-liquid layer

For the electronic structure of quasi-liquid layer, layer-by-layer DOS were calculated with a large number of unoccupied bands (~ 8000) using Vienna ab initio simulation package[56]. The structure was taken from the ab initio MD simulation at 0.45 ps. Due to the large cell size of the slab, only Γ point was used for Brillouin-zone integration. The energy cutoff for plane wave expansion was set to 400 eV. To compare the DOS with experimental O-K edge spectra of ZnO, the main peak position of unoccupied oxygen DOS was aligned to that of experimental O-K edge spectra.

In addition, HSE hybrid functional calculation was carried out to confirm the metallic character of the quasi-liquid layer (Supplementary Fig. 19). Due to the large computational cost, the energy cutoff for plane wave expansion was set to 320 eV and only Γ point was used for Brillouin-zone integration.

## Data availability

Relevant data supporting the key findings of this study are available within the article and the Supplementary Information file. The theoretical atomic structure data generated in this study have been deposited within this article under VASP and XYZ atomic structure formats. All other raw data generated during the current study are available from the corresponding authors upon request. Source data are provided with this paper.

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

## Acknowledgements

This work was supported by the Samsung Research Funding & Incuba-tion Center of Samsung Electronics under Project Number SRFC-MA1702-01 and also partly by the National Research Foundation of Korea (NRF) funded by the Korea government (MSIT) (No. NRF-2020R1A2C2101735, No. NRF-2018R1A2B6004394, No. NRF-2018R1D1A1B07044564). This work was supported by Advanced Facility Center for Quantum Technology, Sungkyunkwan University (SKKU).

## Author contributions

S.H.O. conceived this study. Z.W., J.C.K., and H.Y.J. prepared the in-situ heating samples. Z.W., S.B.L and S.H.O. conducted in-situ TEM heating experiment. Z.W. and B.S.P. conducted STEM EDS and EELS analysis. Z.W. and J.J.S. conducted image processing and analyzing. J.B. con-ducted DFT calculation. J.B. and J.L. conducted MD calculations. All authors contributed to interpretation of data and visualization of results. and Z.W., S.H.O. prepared the manuscript and all authors reviewed and edited it.

## Competing interests

The authors declare no competing interests.
