## [Peer Review File · Nature Communications]

REVIEWER COMMENTS

Reviewer #1 (Remarks to the Author):

The work "Vacancy driven surface disorder catalyzes anisotropic evaporation of polar ZnO(0001) surface", by Z. Wang et al. combines high-resolution transmission electron microscopy (HRTEM) with computations (DFT and Car-Parinello MD) to investigate and rationalize the atomic-scale mechanisms driving the anisotropic evaporation of ZnO, which occur preferentially along the polar [0001] direction.

The work is well motivated and is put into the context of previous, related research. It is very thorough in its experimental and theoretical investigations and provides high-quality and convincing data sets. The main result is unveiling the origin of the anomalous evaporation of ZnO, which, as is convincingly argued, is ultimately due to the polar nature of its (0001) termination. The insights gathered by the authors shed new light on compensation mechanisms at work at polar surfaces, and they may be useful to understand better the growth of ZnO nanostructures. We recommend publication in Nature Communications provided that the authors satisfactorily address the issues listed below.

1) The presentation needs revising. The amount of data provided by the authors is impressive yet not well-structured. The abstract, the introduction, and the conclusions are very clear and nicely summarize the contents. On the other hand, the results and discussion are dispersive and, at times, hard to follow. One loses track of the main message/results when reading through the details of each set of results. To improve readability, we recommend the following:

- a. Inserting an outline at the end of the introduction to guide the reader through the work.
- b. Better introducing and motivating each set of results. Having a summary for each set of results may help.
- c. Better separating paragraphs (e.g., the authors could have an additional paragraph at the beginning of page 10, something like: "desorption energies and vacancy diffusion").
- d. Merge the "extended data" and the supplementary data. Having two types of supporting information is confusing.

We also recommend that the authors restructure the paper to avoid scattered discussions on the same issue (e.g., considerations about beam damage, found on page 8 and page 11; desorption of O, found at page 11 and page 18, etc.).

2) To determine the desorption energies in DFT, single atoms are moved away from the slab and the corresponding energy is determined. This is different from what is normally done – where a slab with a missing atom is referred to a reference (here an O₂ molecule or metallic Zn). While this standard approach often raises the problem or the reference, it avoids treating single atoms with DFT. Also, this unusual approach makes it hard to compare to other works. How do the calculated desorption energies compare with the more standard approach?

3) The authors claim that the quasi-liquid layer is metallic (Extended data 10) based on the DFT-calculated DOS. However, DFT is not always trustworthy when determining bandgaps. Can the authors comment on this? We feel that the metallicity of the quasi-liquid is not well discussed in the context of the other results. Do the authors think that the metallicity is the "ultimate" mechanism that allows the system to compensate the polarity through charge screening? In other words, do the authors think that vacancy diffusion and dissociative desorption are driven by the need to form a metallic layer that is able to compensate polarity?

4) Minor errors:

- a. Numbering of the movies wrong - all video 6?? File names of the movies?
- b. Pag. 3 line 8: staking \diamond stacking
- c. Pag. 10: ALBe, not ABLe
- d. Pag. 10 line 12: "these DFT results rationalize *THAT* only the Zn atoms can preferentially desorb from the polar (0001) surface" (*THAT* missing?)

Reviewer #2 (Remarks to the Author):

This work studies the Zn-terminated ZnO (0001) surface, with a focus on the dynamic processes of the solid-vapor interface during evaporation. It has been established before that there is an important role of the surface disorder and vacancies in stabilising a polar oxide surface; the current study shows that these vacancies help the dissociative evaporation process of the polar ZnO (0001) surface. The latter with the use of both surface analysis experimental techniques as well as (MD) ab-initio computational calculations. In general, the results look promising and I do recommend this paper for publication in Nature Communications after the following comments/questions are addressed.

Main concerns: make the discussion on the stability of ZnO polar surfaces stronger and improve the DFT (and AIMD) details given in the computational section. Authors jump for one to other method, with, apparently, no reason. Surface dipole needs to be cancelled out to make realistic ZnO polar surface models; otherwise, the models are not realistic.

In the abstract (and summary), one has to be clear that the dissociative evaporation is at +300 C.

Define VZnS

There are a wide number of surface features in both Zn and O-terminated ZnO polar surfaces. The triangular pits on the O-terminated is only one of them, hexagonal patterns also appear on experiment, stripes, even (1x1) surfaces with no evidence of surface reconstruction. For pristine ZnO polar surfaces, studies (10.1038/nmat4804, 10.1021/acs.chemmater.7b01487) have shown that disorder and its entropic factors are the ones responsible for their stability. I would recommend a more thorough discussion on the stability of the ZnO polar surfaces, as this is the first step to understand the surface nonstoichiometry, which is needed for stabilising the dipole. For example, nowhere is mentioned what is the amount of Zn(O) vacancies to stabilise the polar surfaces.

What is the evaporation behaviour of the O-terminated polar face?

Why a higher evaporation rate usually means higher growth rate?

"it is not surprising that the polar ZnO (0001) surface has a higher growth rate than all the other surface orientations¹⁵⁻¹⁸." – There are experiment reports (10.3938/jkps.56.498) and calculations (10.1021/acs.chemmater.7b01487) that show that the (000-1) surface is more stable than its Zn-terminated counterpart. Under thermodynamic equilibrium, the most stable ZnO surface is the nonpolar (10-10) surface, which takes most of the surface area of the ZnO crystal.

The entropic factors play a key role on both polar surfaces, not only on the Zn-terminated one. And even in other ionic materials, such as CeO₂ (10.1038/nmat4804)

There is no such a thing as a pristine stoichiometric ZnO polar surface. They are not stable. Surface disorder and vacancies are needed to compensate the dipole and lower their surface energy. Vacancies are intrinsic features in ZnO polar surfaces. In the manuscript, this has to be clearer.

T=300 C is related to the desorption energies of Zn and/or O shown in Figure 2d?

What is the desorption energy for O (Zn) atoms on the O-terminated surface?

Define SV.

The Arrhenius plot of the moving rate measured at various temperatures yields the activation energy of 0.34 eV (Extended Data Fig. 1c), which is much smaller than the reported value (~2.9-3.7 eV) for ZnO (s) = Zn (g) + 1/2 O₂ (g)³⁰ – why?

What is the Zn:O ratio for as-cleaved (0001) ZnO surface?

In the model, were Zn vacancies on the (0001) to compensate for the dipole included? How?

"It is interesting to note that the displacement vectors, although fluctuate constantly with time, point more frequently the in-plane directions." – what is the reason for this?

What about the dipole in your calculations? Is it cancelled?

"The layer-by-layer DOS shows that the disordered layer is metallic, so that its free electrons can screen the surface charge of the (0001) polar surface" – there is no experimental evidence that on clean ZnO polar surfaces exist surface conduction (metallisation), see 10.1021/acs.chemmater.7b01487 and references therein. Usually, this mechanism leads to a high (unstable) energy surfaces.

I think that is important to link these results to previous ones where it has been exposed that surface disorder and vacancies are the key for stabilising the ZnO polar surfaces.

It is not clear to me. Please, clarify how the surface models were built. Did you run calculations on clean stoichiometric ZnO polar surfaces? If so, those models do not represent the ZnO polar surfaces since there is a big dipole across the c axis and surface energies are large (unstable), compared to the other ZnO surfaces. The first step to model ionic polar surfaces is to diminish as much as possible the dipole. Considering the formal charges of the Zn and O species, there should be an ionic transfer of ~24% Zn²⁺ of the (0001) surface to the (000-1) surface to cancel out the dipole. This agrees with the best fit to X-ray diffraction data (10.1016/S0169-4332(00)00172-0), which was obtained with 25% Zn²⁺ vacancies at the Zn terminated surface.

"We generated three samples with different vacancy distribution for each Zn deficient case, except the pure ZnO, and the calculated elastic constants for the samples are averaged." –what did you use the pure ZnO models for? What is the different vacancy distribution? Did you transfer those ionic charges to the O-terminated surface or just created the vacancies?

What is the reason for using two different functionals (LDA+U and PBE)? Why not to use either or to be more consistent? For example, on the latter, there is no account for the on-site Coulomb energy. I understand that CP2K could provide faster calculations for the AIMD simulations, then, one could use CP2K and PBE+U (or hybrids, not that expensive anymore on CP2K) and be more consistent with the results. After the AIMD simulation with CP2K+PBE the authors go back to VASP (with LDA+U or PBE? – it is not clear in the text). Calculations need to be consistent to provide an easier and fair comparison among them.

Reviewer #3 (Remarks to the Author):

The manuscript deals with the study of ZnO materials and how they equilibrate when cut through a polar surface. The work is carefully done and presents important implications. However, I have severe reserves on the modeling part.

1. The use of LDA+U is really not justified, better functionals can be employed for this purpose and since the semiconductor nature of the native material is so relevant I think that either severe benchmarking or a better functional even including hybrid methods are needed to certify the relevance of the values obtained.

2. The nature of the diffuse layer formed is not seen in the computational part, can diffusion and aggregation be computed there?

3. Some of the energies are far to large to be representative. The computational part needs to

indicate the reference energies employed.

4. The structures need to be uploaded to a repository for other scientist to reuse them

Reviewer #1

The work "Vacancy driven surface disorder catalyzes anisotropic evaporation of polar ZnO(0001) surface", by Z. Wang et al. combines high-resolution transmission electron microscopy (HRTEM) with computations (DFT and Car-Parinello MD) to investigate and rationalize the atomic-scale mechanisms driving the anisotropic evaporation of ZnO, which occur preferentially along the polar [0001] direction. The work is well motivated and is put into the context of previous, related research. It is very thorough in its experimental and theoretical investigations and provides high-quality and convincing data sets. The main result is unveiling the origin of the anomalous evaporation of ZnO, which, as is convincingly argued, is ultimately due to the polar nature of its (0001) termination. The insights gathered by the authors shed new light on compensation mechanisms at work at polar surfaces, and they may be useful to understand better the growth of ZnO nanostructures. We recommend publication in Nature Communications provided that the authors satisfactorily address the issues listed below.

Reply: We thank the reviewer for the constructive and insightful comments that immensely helped us to improve the manuscript. Our detailed point-by-point responses to the questions and comments are appended below.

Comment 1: The presentation needs revising. The amount of data provided by the authors is impressive yet not well-structured. The abstract, the introduction, and the conclusions are very clear and nicely summarize the contents. On the other hand, the results and discussion are dispersive and, at times, hard to follow. One loses track of the main message/results when reading through the details of each set of results. To improve readability, we recommend the following:

Reply: We appreciate the reviewer's feedback and suggestion. To improve the readability of manuscript, we restructured and revised the manuscript following the suggestions as much as we can.

a. Inserting an outline at the end of the introduction to guide the reader through the work.

Reply: The last paragraph of the introduction in the original manuscript was prepared as an outline summarizing the key findings of our work. We revised this paragraph to guide the reader through the work.

b. Better introducing and motivating each set of results. Having a summary for each set of results may help.

Reply: We provided a short summary for each set of results to better introduce and motivate each set of results.

c. Better separating paragraphs (e.g., the authors could have an additional paragraph at the beginning of page 10, something like: "desorption energies and vacancy diffusion").

Reply: We separated the corresponding paragraph and added the sub-heading "Anisotropic desorption energies and vacancy diffusion".

d. Merge the "extended data" and the supplementary data. Having two types of

supporting information is confusing.

We also recommend that the authors restructure the paper to avoid scattered discussions on the same issue (e.g., considerations about beam damage, found on page 8 and page 11; desorption of O, found at page 11 and page 18, etc.).”

Reply: We merged “extended data” with “supplementary data”. Now we have a single supplementary information. We restructured the manuscript to avoid scattered discussions on the same issue, especially the electron beam effects (we moved some part of description to the figure caption to minimize the distraction of readership).

Comment 2: To determine the desorption energies in DFT, single atoms are moved away from the slab and the corresponding energy is determined. This is different from what is normally done – where a slab with a missing atom is referred to a reference (here an O₂ molecule or metallic Zn). While this standard approach often raises the problem or the reference, it avoids treating single atoms with DFT. Also, this unusual approach makes it hard to compare to other works. How do the calculated desorption energies compare with the more standard approach?”

Reply: We appreciate the reviewer for this comment. The values in the manuscript are actually the “desorption barriers” for the desorption processes. The reason that we used the desorption barrier is that we considered the kinetic process of the Zn or O desorption. Note that the same method had been used in a previous work [R1], where the kinetics of absorption and desorption of hydrogen on graphite had been studied. We admit that this is misleading, and for better comparison to other works, as the reviewer mentioned, we have changed the values (desorption barriers) to “desorption energy”.

For your information, we describe how the desorption energy is related to the corresponding desorption barrier below:

Because there is no uphill in the energy landscapes of the desorption processes [See Supplementary Fig. S8], the desorption barrier (DB) of an atom X, $E_{DB}(X)$, was calculated by $E_{DB}(X) = E(S:V_X + X^{away}) - E(S)$, where $E(S)$ and $E(S:V_X + X^{away})$ are the total energies of the pure surface and the surface containing a X vacancy (V_X) with a X atom far away from the surface, respectively. Here, $E(S:V_X + X^{away})$ is approximately the sum of the total energies of the surface with V_X $E(S:V_X)$ and the isolated X atom $E(X)$, i.e., $E(S:V_X + X^{away}) \cong E(S:V_X) + E(X)$, because the X atom is far away from the surface and behaves as an isolated one. In our calculations, the energy differences between the left- and right-hand sides are below or about 0.1 eV for all the cases. On the other hand, the desorption energy (DE) of the X atom, $E_{DE}(X)$, can be calculated by $E_{DE}(X) = E(S:V_X) + E_{ref}(X) - E(S)$, where $E_{ref}(X)$ is the reference energy of the X atom, i.e., $E_{ref}(Zn)$ and $E_{ref}(O)$ are the single atom energies of the Zn metal and O₂ molecule, respectively. Thus, the difference $E_{DB}(X) - E_{DE}(X) \cong E(X) - E_{ref}(X) = \Delta E(X)$, which is the energy change of a Zn or O atom associated with the formation of a Zn metal or O₂ molecule. In short, the desorption energies $E_{DE}(X)$ are smaller than the corresponding desorption barriers $E_{DB}(X)$ by the energy change $\Delta E(X)$, i.e., $E_{DE}(X) \cong E_{DB}(X) - \Delta E(X)$. In our calculations, $\Delta E(Zn)=1.8$ eV and $\Delta E(O)=3.3$ eV. We have included this discussion in the caption of Supplementary Fig. S8.

Reference:

R1. Hornekaer, L. *et al.* Clustering of Chemisorbed H(D) Atoms on the Graphite (0001) Surface due to Preferential Sticking. *Phys. Rev. Lett.* **97**, 186102 (2006).

Comment 3: The authors claim that the quasi-liquid layer is metallic (Extended data 10) based on the DFT-calculated DOS. However, DFT is not always trustworthy when determining bandgaps. Can the authors comment on this?

Reply: As the reviewer pointed out, it is well known that standard DFT calculation frequently suffers from a band gap underestimation problem. The hybrid functionals (such as HSE) are used to remedy the weaknesses of standard DFT. So, we performed the HSE calculations on the structure used for standard DFT calculation, [shown in Fig. R1 (a)], consisting of disordered region and ordered layers. We find that disordered region still remains metallic even with HSE calculations [Fig. R1 (c)], supporting that disordered region is indeed metallic, which is consistent with the standard DFT results [Fig. R1(b)]. We included this results in the supplementary information as Supplementary Fig. S19.

Fig. R1. (a) Snapshot of *ab initio* MD simulation result at 0.45 ps. Layer-by-layer density of states computed with (b) standard PBE and (c) HSE functionals.

Comment 4: We feel that the metallicity of the quasi-liquid is not well discussed in the context of the other results. Do the authors think that the metallicity is the “ultimate” mechanism that allows the system to compensate the polarity through charge screening? In other words, do the authors think that vacancy diffusion and dissociative desorption are driven by the need to form a metallic layer that is able to compensate polarity?”

Reply: We appreciate the reviewer for this insightful question. As to the polarity compensation of the ZnO (0001) surface, the metallicity of quasi-liquid certainly works well in screening the surface charges. However, this should not be regarded as the “ultimate mechanism” for the polarity compensation. As shown in Figure R2 (Figure 1 in the revised manuscript), a pristine ZnO (0001) polar surface can be

stabilized readily by forming Zn vacancies and/or atomic steps at temperatures below 300 °C without lattice disordering or quasi-liquid layer formation. The observed stabilization mechanism indeed agrees well with those reported in the previous studies. The formation of quasi-liquid should be considered as a “result” of the growing Zn deficiency with temperature due to the preferential desorption of Zn and vacancy diffusion which are thermally activated processes driven by the polarity compensation.

Fig. R2. HRTEM image of a pristine ZnO (0001) surface and quantitative analysis of the intensity of Zn atomic columns. (a) HRTEM image of ZnO crystal recorded at a NCSI condition. The Zn-terminated (0001) surface in edge-on orientation is indicated by white arrow. (b) Magnified view of the region outlined by white lines in (a). The region was selected for the quantitative analysis of the intensity of Zn atomic columns to assess the number of Zn atoms. (c) Normalized intensity of the Zn atoms displayed in 2D map for the individual Zn columns and histogram for the layer-averaged values with standard deviation. Atomic model is overlaid to guide the position of Zn atoms.

Even though not the ultimate mechanism for the polarity compensation, the quasi-liquid influences the surface relaxation of the ZnO (0001) surface. Note that the surface relaxation of a polar surface varies sensitively depending on the way how the polarity is compensated. Considering the surface state of ZnO (0001) at the condition where quasi-liquid evolves, the (0001) surface becomes rough and dynamically evolving due to the dissociative evaporation, so that the surface polar termination cannot be defined as an ideal flat Zn layer. Furthermore, the V_{Zn} diffuses away from the surface toward the bulk, resulting in an extended distribution. As a consequence, the surface relaxation (the interlayer spacing, in specific) and the ionic polarization of the ZnO (0001) covered by quasi-liquid are modified noticeably compared to the pristine surface; for example, the interlayer relaxation, once strongly confined to the first layer of pristine (0001) surface is extended over multiple layers from the surface due to the extended distribution of Zn vacancies and the ionic polarization decreases and disappears near the surface.

Fig. R3. Surface relaxation of the ZnO (0001) polar surface stabilized by V_{Zn} . (a) HRTEM image of ZnO crystal recorded at a NCSI condition. (b) Magnified view of the region outlined by white lines in (a). The region was selected for the measurement of the interlayer spacing of Zn-O bilayer, d_{ij} . (c) Change of the interlayer spacing from the surface to the bulk. The first Zn-O bilayer, d_{12} , shows a large contraction up to ~30%, which agrees well the atomic simulation reported in the literature [R2, R3].

In the current manuscript we did not discuss the quasi-liquid in detail as we have been considering this as a subject for separate paper (please refer to Figure R3). Instead, we focus more on the quasi-liquid as a self-catalyst promoting the fast-anisotropic evaporation (and crystal growth) of ZnO nanostructures. For more comprehensive description of the effects of quasi-liquid, we have added the following sentences.

The layer-by-layer DOS shows that the quasi-liquid is metallic, so that in principle its free electrons can screen the surface charges of the (0001) polar surface (Fig. 6d). However, the metallicity of quasi-liquid should not be regarded as the ultimate mechanism for the polarity compensation. As shown in Fig. 1, a pristine ZnO (0001) polar surface can be stabilized readily by forming V_{Zn} and/or atomic steps at temperatures below 300 °C without lattice disordering or quasi-liquid layer formation. The formation of quasi-liquid should be considered as a result of the growing Zn deficiency with temperature due to the preferential desorption of Zn and vacancy diffusion which are thermally activated processes driven by the polarity compensation. Considering the surface state of ZnO (0001) at the condition where quasi-liquid evolves, the surface polar termination cannot be defined by an ideal Zn atomic layer as the surface roughness is constantly evolving during the dissociative evaporation. Furthermore, the V_{Zn} diffuses away from the surface toward the bulk, resulting in an extended distribution of V_{Zn} . As a consequence, the surface relaxation (the interlayer spacing, in specific) and the ionic polarization of the ZnO (0001) covered by quasi-liquid are modified noticeably compared to the pristine surface; the interlayer relaxation, once strongly confined to the first layer of pristine (0001) surface, is extended over multiple layers from the surface due to the extended distribution of Zn vacancies and the ionic polarization decreases and disappears near the surface (data not shown).

Reference:

R2. Meyer, B. & Marx, D. Density-functional study of the structure and stability of ZnO surfaces. *Phys. Rev. B* **67**, 035403 (2003).

R3. Kresse, G., Dulub, O. & Diebold, U. Competing stabilization mechanism for the polar ZnO(0001)-Zn surface. *Phys. Rev. B* **68**, 245409 (2003).

Comment 4: Minor errors:

- a. Numbering of the movies wrong - all video 6?? File names of the movies?
- b. Pag. 3 line 8: staking \diamond stacking
- c. Pag. 10: ALBe, not ABLe
- d. Pag. 10 line 12: "these DFT results rationalize *THAT* only the Zn atoms can preferentially desorb from the polar (0001) surface" (*THAT* missing?)

Reply: We appreciate the reviewer for very careful assessment of the manuscript. Following the comments, we have corrected all the errors.

Reviewer #2

This work studies the Zn-terminated ZnO (0001) surface, with a focus on the dynamic processes of the solid-vapor interface during evaporation. It has been established before that there is an important role of the surface disorder and vacancies in stabilising a polar oxide surface; the current study shows that these vacancies help the dissociative evaporation process of the polar ZnO (0001) surface. The latter with the use of both surface analysis experimental techniques as well as (MD) ab-initio computational calculations. In general, the results look promising and I do recommend this paper for publication in Nature Communications after the following comments/questions are addressed.

Reply: We thank the reviewer for the constructive and insightful comments that immensely helped us to improve the manuscript. Our detailed responses to the questions and comments are appended below.

Main concerns: make the discussion on the stability of ZnO polar surfaces stronger and improve the DFT (and AIMD) details given in the computational section. Authors jump for one to other method, with, apparently, no reason. Surface dipole needs to be cancelled out to make realistic ZnO polar surface models; otherwise, the models are not realistic.”

Reply: Following the reviewer’s suggestions, we strengthened the discussion of ZnO polar surface, improved the DFT details, and provided the detailed description on how realistic ZnO polar surface models were constructed with cancellation of surface dipole.

Comment 1: “In the abstract (and summary), one has to be clear that the dissociative evaporation is at +300 C.”

Reply: Following the comment, we have included the temperature (300 °C) in the abstract and summary.

Comment 2: “Define V_{ZnS} ”

Reply: We used V_{Zn} as the abbreviation of Zn vacancy, and V_{Zns} is the plural of V_{Zn} , i.e., Zn vacancies. However, to avoid confusion, we used V_{Zn} throughout the manuscript

Comment 3: “There are a wide number of surface features in both Zn and O-terminated ZnO polar surfaces. The triangular pits on the O-terminated is only one of them, hexagonal patterns also appear on experiment, stripes, even (1x1) surfaces with no evidence of surface reconstruction. For pristine ZnO polar surfaces, studies (10.1038/nmat4804, 10.1021/acs.chemmater.7b01487) have shown that disorder and its entropic factors are the ones responsible for their stability. I would recommend a more thorough discussion on the stability of the ZnO polar surfaces, as this is the first step to understand the surface nonstoichiometry, which is needed for stabilising the dipole. For example, nowhere is mentioned what is the amount of

Zn(O) vacancies to stabilise the polar surfaces.”

Reply: Following the reviewer’s suggestion, we added more information on the stability of ZnO polar surfaces, such as the amount of V_{Zn} needed for stabilizing the Zn-terminated surface (one vacancy per every surface four unit cell) and the importance of the entropic factors to rationalize the nonstoichiometric and disordered nature of the ZnO polar surfaces.

On page 3: It has been shown that Zn vacancy (V_{Zn}) plays a central role in stabilizing a pristine ZnO (0001) polar surface; a simple ionic model predicts that one Zn vacancy (V_{Zn}) in every four surface unit cells can stabilize the ZnO (0001) polar surface^{2,8}. Various surface features have been reported for the ZnO (0001) polar surfaces, which are closely related to the formation of V_{Zn} ^{6,9}. For example, when annealed in vacuum, the ZnO (0001) polar surface tends to exhibit triangular or hexagonal pits of V_{Zn} with exposing O-terminated step edges, which is favored by the Madelung energy^{6,10}. It has been further shown that the regular array of these steps to a specific vicinal surface orientation, such as the (10-14), can compensate the dipole moment along the surface normal^{7,11}. A recent study has well documented the stability of ZnO polar surfaces, which is critically governed by the entropic factors and associated structural disorder¹². This implies that ZnO polar surfaces are nonstoichiometric and intrinsically rough at the atomic scale.

On page 16: As shown in Fig. 1, a pristine ZnO (0001) polar surface can be stabilized readily by forming Zn vacancies and/or atomic steps at temperatures below 300 °C without lattice disordering or quasi-liquid layer formation.

However, considering the limited space, we could not add further detailed discussion on this subject as we like to stay focused on the main point of this manuscript, which is the anisotropic evaporation of ZnO (0001) polar surface due to the surface disorder at elevated temperatures (> 300 °C).

Comment 4: “What is the evaporation behaviour of the O-terminated polar face?”

Reply: We have also conducted the in-situ heating experiments on the O-terminated (000-1) polar surface under the same condition employed for the Zn-terminated surface. First, as already shown in Fig. 2d, Supplementary Fig. S5d. and Movie. S4, there is no quasi-liquid layer formed on the (000-1) surface, as well as other (10-11), (10-10), (-1011) surfaces; the quasi-liquid formed only on the (0001)-Zn polar surface. Second, we have traced the position of the O-terminated surface with time during the evaporation. The plot is shown below in Fig. R4(b). It clearly shows that the surface move at a almost constant and relatively low rate, which is similar to that of (10-11), (10-10), (-1011) surfaces.

Fig. R4. (a) Time series HRTEM images of polar (000-1) surface edge recorded at 300 °C. The surface edge was outlined in different color to trace its movement during evaporation. (b) Plot of the moving distance of the (000-1) surface orientation during evaporation.

Comment 5: “Why a higher evaporation rate usually means higher growth rate?”

Reply: We should admit that this statement is only valid for a reversible growth and evaporation process under conditions of thermodynamic equilibrium. Strictly speaking, most practical growth processes of ZnO nanostructures take place under non-equilibrium conditions. Moreover, the vapor sources used for the growth are different from the evaporating species from ZnO surfaces. Nonetheless, we made this statement based on the known fact that both growth and evaporation of ZnO nanostructures are fast along the [0001] direction. The sentence has been modified to convey the correct message in page 4, which reads now. **It is also well known that ZnO nanostructures grow abnormally fast along the [0001] direction¹⁸⁻²¹.**

Comment 6: “it is not surprising that the polar ZnO (0001) surface has a higher growth rate than all the other surface orientations¹⁵⁻¹⁸.” – There are experiment reports (10.3938/jkps.56.498) and calculations (10.1021/acs.chemmater.7b01487) that show that the (000-1) surface is more stable than its Zn-terminated counterpart. Under thermodynamic equilibrium, the most stable ZnO surface is the nonpolar (10-10) surface, which takes most of the surface area of the ZnO crystal.”

Reply: The surface with a higher surface energy usually has a higher growth rate, which will reduce its surface area, thereby decrease the total surface energy of whole system. As reported in literature, the (000-1) surface is more stable than its Zn-terminated (0001) counterpart. Overall, the nonpolar (10-10) is the most stable surface. The equilibrium shape of ZnO predicted by Wulff construction shows that the (0001) surface shows the smallest area than the all other surfaces [R4]. This

strongly anisotropic surface energy explains why ZnO grows easily into nanowires along the [0001] direction. We have modified the sentence to read: We note that the surface energy of the (0001) polar surface is the largest among other low-index surface orientations of ZnO²²⁻²⁴. The surface with a higher surface energy usually has a higher growth rate, which will reduce its surface area, thereby decrease the total surface energy of whole system²⁵.

We have added the following reference:

R4. Wilson, H. F., Tang, C. & Barnard, A. S. Morphology of zinc oxide nanoparticles and nanowires: Role of surface and edge energies. *J. Phys. Chem. C* **120**, 9498–9505 (2016).

Comment 7: “The entropic factors play a key role on both polar surfaces, not only on the Zn-terminated one. And even in other ionic materials, such as CeO₂ (10.1038/nmat4804)”

Reply: Yes, the reviewer is absolutely right. The entropic factors play a key role on both polar surfaces at high temperatures. Similar to the Zn-terminated surface, the O-terminated surface is stabilized via non-stoichiometric surface reconstruction involving the formation of oxygen vacancies. However, different from the Zn-terminated surface, the surface disorder does not result in the formation of quasi-liquid on the O-terminated surface. In order to validate the O-terminated surface, we performed the *ab initio* MD simulation on the O-terminated surface with surface oxygen vacancies in the same number as in Zn vacancies of Zn-terminated surface. As shown in Fig. R5, surface disorder on the O-terminated surface is occurred which is similar to CeO₂ [R5]. However, the Zn atoms do not evaporate on O-terminated surface as opposed to evaporation of oxygen molecules on the Zn-terminated surface. This is due to the different energetics of desorption energy and lack of an oxygen deficient solid phase in the Zn-O binary system. As a result, the remaining Zn atoms on O-terminated surface maintain the lattice. We have included the discussion on page 17, to read “However, different from the (0001), the surface disorder at high temperature does not result in the formation of quasi-liquid on (000-1) polar surface (Supplementary Fig. S21). This is due to the different desorption behavior of Zn atoms from the (000-1) surface and the lack of an oxygen deficient solid phase in the Zn-O binary system.” The Fig. R5 has been added to the Supplementary Figure (Supplementary Fig. S21).

Reference:

R5. Capdevila-Cortada, M. & López, N. Entropic contributions enhance polarity compensation for CeO₂ (100) surfaces. *Nat. Mater.* **16**, 328–334 (2017).

Fig. R5. Surface disorder of vacancy-stabilized ZnO polar surfaces. Relative energy of (a) Zn-terminated (0001) surface with V_{Zn} and (b), O-terminated surface with oxygen vacancy as a function of time during MD simulation. Snapshots at 0.24, 0.45 and 1.6 ps are shown in the inset of figure. Quasi-liquid like disordered layer forms only on the (0001) surface.

Comment 8: “There is no such a thing as a pristine stoichiometric ZnO polar surface. They are not stable. Surface disorder and vacancies are needed to compensate the dipole and lower their surface energy. Vacancies are intrinsic features in ZnO polar surfaces. In the manuscript, this has to be clearer.”

Reply: The reviewer is absolutely right. We have made this point clearer in the revised manuscript by adding a new section with the title of “Stabilization of ZnO (0001) surface by Zn vacancies” and with a new figure (Fig. R2 below, Fig. 1 in the revised manuscript). Figure R2 now explicitly shows the existence of V_{Zn} in the pristine (0001) polar surface at room temperature, which agrees well with those reported in the literatures.

Fig. R2. HRTEM image of a pristine ZnO (0001) surface and quantitative analysis of the intensity of Zn atomic columns. (a) HRTEM image of ZnO crystal recorded at a NCSI condition. The Zn-terminated (0001) surface in edge-on orientation is indicated by white arrow. (b) Magnified view of the region outlined by white lines in (a). The region was selected for the quantitative analysis of the intensity of Zn atomic columns to assess the number of Zn atoms. (c) Normalized intensity of the Zn atoms displayed in 2D map for the individual Zn columns and histogram for the layer-averaged values with standard deviation. Atomic model is overlaid to guide the position of Zn atoms.

Comment 9: “T=300 C is related to the desorption energies of Zn and/or O shown in Figure 2d?”

Reply: The desorption energies is calculated by DFT calculations, which assume that systems are in the zero temperature. However, it is known that the desorption energies remain nearly the same, although the system temperature increases to 300 °C. We added this information to the figure caption of Figure 3d in the revised manuscript, which read: **The desorption energies calculated by DFT at 0 K.**

Comment 10: “What is the desorption energy for O (Zn) atoms on the O-terminated surface?”

Reply: The calculated desorption energies for O and Zn atoms on the O-terminated polar surface are 1.2 and 6.2 eV, respectively. The O desorption energy is larger than the Zn desorption energy (0.3 eV) on the (0001) surface. This may be the reason for the observed slow evaporation rate on the O-terminated surface (Fig. R4, Supplementary Fig. S5d). Regarding this comment, we have included the sentence on page 9, to read: **In addition, we also considered the O-terminated (000-1) polar surface and found that the calculated desorption energies for O and Zn atoms are 1.2 and 6.2 eV, respectively. The O desorption energy of the O-terminated (000-1) polar surface is larger than the Zn desorption energy (0.3 eV) of the (0001) polar surface.**

Comment 11: “Define SV”.

Reply: We have defined SV as the abbreviation of “solid-vapor” on page 5.

Comment 12: “The Arrhenius plot of the moving rate measured at various temperatures yields the activation energy of 0.34 eV (Extended Data Fig. 1c), which is much smaller than the reported value (~2.9-3.7 eV) for $\text{ZnO (s)} = \text{Zn (g)} + \frac{1}{2} \text{O}_2 \text{(g)}$ – why?”

Reply: The reaction energy (~2.9-3.7 eV) mentioned in the manuscript corresponds to the calculated energy based on the thermodynamic data (Gibbs free energy) of bulk phases. In this calculation the additional energy terms, such as the surface energy and the electrostatic energy which are strongly anisotropic, are not considered. As we already stated in the manuscript, the evaporation rate of ZnO predicted by such bulk free energies cannot account for the experimentally measured rate. Because of this reason many researchers postulated the existence of a “non-equilibrium” condition, where the evaporation is governed by the diffusion of defects or interface reactions. For example, the formation of an impervious layer of non-stoichiometric on the (0001) surface has been postulated [R6] (Ref. 14 in the revised manuscript), so that the evaporation is essentially governed by the corresponding oxide, not by ZnO crystal. In summary, the experimentally measured evaporation rate of the (0001) polar surface cannot be accounted for by the reaction energy calculated by thermodynamic data and its activation energy must be smaller than the thermodynamic prediction.

Judging from the $t^{1/2}$ dependency of the moving rate, it is clear that the evaporation is controlled by a diffusion process when quasi-liquid is formed. From the Arrhenius plot of the evaporation rate the activation energy is measured to be ~0.34 eV. From these two results, we can conclude that the evaporation is controlled by the “diffusion process with the activation energy of 0.34 eV” when quasi-liquid is formed. This energy barrier is compared well with the diffusion barrier of Zn vacancy (0.77 eV); once the surface Zn atoms are desorbed, then the inward diffusion of V_{Zn} becomes the rate-limiting step of evaporation but this can be easily activated at 300 °C due to the low activation energy (0.77 eV). The smaller measured activation energy (0.34 eV) than the calculated one (0.77 eV) is attributable to the catalytic effects of quasi-liquid and/or the electron beam stimulation (Supplementary Note 3). However, the electron beam irradiation is not the major cause of the formation of quasi-liquid and subsequent accelerated evaporation of ZnO (0001) polar surface as we proved by the control experiment (Supplementary Figs. S1, S9a, S10) and the calculation (Supplementary Fig. S9b) showing no significant influence on the surface evaporation process.

Reference:

R6. Secco, E. A. DECOMPOSITION OF ZINC OXIDE. *Can. J. Chem.* **38**, 596–601 (1960).

Comment 13: “What is the Zn:O ratio for as-cleaved (0001) ZnO surface?”

Reply: We used a single crystal ZnO of which one of the (0001) sides being polished into an optical grade (CrysTec GmbH, Germany). When prepared to a cross-sectional TEM sample by the Ga^+ ion beam milling process, however, the pristine (0001) surface is thinned away and fresh (0001) surface edges are exposed near the thinnest part of the TEM sample. This surface remains clean in vacuum in the

absence of polar adsorbates, and, thus, can be regarded as a stabilized (0001) polar surface. The quantitative analysis of HRTEM image revealed that the occupancy of Zn columns at the surface is lower than that of oxygen columns (Fig. 1), demonstrating that V_{Zn} formed and play a central role in the stabilization of (0001) polar surface.

Comment 14: “In the model, were Zn vacancies on the (0001) to compensate for the dipole included? How?”

Reply: First of all, we agree with the reviewer that a pristine (0001) polar surface is unstable due to the dipole and must be stabilized by Zn vacancies, which we confirmed experimentally by HRTEM image analysis (as shown in Fig. 1 in the revised manuscript). Apart from this fact, the DFT and MD calculations in this study have been carried out using the atomic models with different extent of Zn vacancies depending on the information we want to know. For example, the desorption energy barriers for the surface Zn and O atoms were calculated by using the pristine surface models without Zn vacancies for each orientation (Here we passivated all surface dangling bonds on the opposite side of the considered surface using pseudo-hydrogen in order to prevent artificial charge transfer and strong internal field; See also Reply of Comment 16). The gained knowledge from these calculations, i.e., the smallest desorption energy barrier for the Zn atom from the (0001) surface, supports the energetic preference of Zn vacancy formation on the (0001) surface. In other words, this is done to prove the reviewer’s thought, i.e., the existence of the Zn vacancies. Except these calculations, Zn vacancies are introduced in all the other calculations, for example, the calculation of the diffusion barrier of Zn vacancies (Fig. 3e) and the MD simulation showing the quasi-liquid formation by surface disorder (Fig. 6).

Comment 15: “It is interesting to note that the displacement vectors, although fluctuate constantly with time, point more frequently the in-plane directions.” – what is the reason for this?”

Reply: This is due to the softening of the lattice stiffness, which is caused by Zn vacancies. From our calculation we found that one of the elastic constants, C_{66} which is the elastic constant associated with the in-plane shear stiffness of the (0001) plane, decreases much faster with increase of V_{Zn} concentration than all other elastic constant tensors (refer to Fig. R6). We have stated the reason in the original manuscript on page 12, which reads:

“Given that the displacement and disordering of ZnO lattice is driven by diffusion and accumulation of V_{Zn} , the lattice instability of ZnO caused by V_{Zn} has been studied by calculating the elastic constants of Zn-deficient $Zn_{1-x}O$ using DFT (Supplementary note 4, Figs. S15, S16). [...] While the elastic constants C_{11} and C_{33} represent stiffness against principal strains, C_{66} and C_{44} reflect the resistance against shear along the in-plane direction on the (0001) plane and the (10-10) plane, respectively. [...] The calculated elastic constants with respect to the Zn deficiency x in $Zn_{1-x}O$ shows that both shear elastic constants C_{66} and C_{44} decrease almost linearly with the Zn deficiency (Fig. 4e). Interestingly, C_{66} associated with the in-plane shear stiffness of the (0001) plane decreases with V_{Zn} more rapidly than C_{44} , governing the

lattice stability of $Zn_{1-x}O$. This shear instability predicts that $Zn_{1-x}O$ softens more easily along the in-plane directions of (0001) plane with growing concentration of V_{Zn} , which is compared well with the experimentally observed in-plane dominant displacement field of Zn atomic columns. From the viewpoint of V_{Zn} diffusion, the inward diffusion of V_{Zn} from the surface via the energetically favored out-of-plane migration path is also expected to cause the in-plane dominant displacement field as it produces the lattice distortion predominantly along the in-plane directions”.

Fig. R6. (a) Calculated elastic constants with respect to Zn deficiency x in $Zn_{1-x}O$. All the constants almost linearly decrease with the Zn deficiency x [Zn_{1-x}O]. (b) Calculated elastic constant C_{44} and C_{66} as a function of zinc deficiency (x) in $Zn_{1-x}O$. C_{66} associated with the in-plane shear stiffness of the (0001) plane decreases with V_{Zn} more rapidly than C_{44} . The extrapolation based on a linear fit predicts $C_{66} = 0$ when $x = 0.25$.

Comment 16: “What about the dipole in your calculations? Is it cancelled?”

Reply: In the slab model for the ZnO (0001) surface, there are two surfaces; the one is the Zn-terminated (0001) polar surface, and the other on the opposite side is the O-terminated (000-1) polar surface. If the charge transfer occurs from Zn polar to O polar surfaces, strong and constant electric field is formed inside ZnO. To prevent the charge transfer, we passivate the dangling bonds on the O polar surface using 0.5 fractionally charged pseudo-hydrogen, which was widely used in previous works [R7, R8]. Similar approaches have also been adopted for ZnO [R9, R10]. Figure R7 shows the xy -plane averaged local potential for the passivated surface. As seen in the figure, the surface dipole field is fully compensated in our slab model. To make this point clear, we have included the discussion in the method.

Fig. R7. The xy -plane averaged local potential of the calculated (0001) polar surface. Our supercell contains 10 layers, and the potential dips occurs at each Zn-O layer. The Zn-terminated surface is located at 21.4 Å. The red dotted line is the running average of the local potential. The flatness of the potential implies the full compensation of the surface dipole field.

References:

- R7. Zhang, S. B. & Wei, S.-H. Surface Energy and the Common Dangling Bond Rule for Semiconductors. *Phys. Rev. Lett.* **92**, 86102 (2004).
- R8. Dreyer, C. E., Janotti, A. & de Walle, C. G. Absolute surface energies of polar and nonpolar planes of GaN. *Phys. Rev. B* **89**, 81305 (2014).
- R9. Xu, H. *et al.* Stabilizing forces acting on ZnO polar surfaces: STM, LEED, and DFT. *Phys. Rev. B - Condens. Matter Mater. Phys.* **89**, 235403 (2014).
- R10. Dag, S., Wang, S. & Wang, L.-W. Large surface dipole moments in ZnO nanorods. *Nano Lett.* **11**, 2348–2352 (2011).

Comment 17: “The layer-by-layer DOS shows that the disordered layer is metallic, so that its free electrons can screen the surface charge of the (0001) polar surface” – there is no experimental evidence that on clean ZnO polar surfaces exist surface conduction (metallisation), see 10.1021/acs.chemmater.7b01487 and references therein. Usually, this mechanism leads to a high (unstable) energy surfaces.”

Reply: We appreciate the reviewer for this knowledgeable comment. As far as the ZnO (0001) polar surface is concerned at a low temperature (> 300 °C), the reviewer’s comment is absolutely right. However, we should make it clear that we are dealing with ZnO polar surface at high temperature (300 °C) where both surface conductivity and evaporation rate increases dramatically. It is a very well-known fact that the conductivity of Zn-terminated ZnO surface increases sharply at around 300 °C together with the evaporation rate [R11] (Ref. 13 of the revised manuscript). The present study reveals that the metallic conductivity and fast evaporation of ZnO (0001) surface at $T > 300$ °C originates from the formation of the Zn-deficient, metallic quasi-liquid on the surface. Furthermore, during the evaporation the ZnO (0001) polar surface continuously evolves and never reach the thermodynamically

stable condition. We are not arguing that the metallization of the polar surface by quasi-liquid layer stabilizes the surface, but it could compensate the dipole and then reduce the evaporation barrier.

Reference:

R11. Kohl, D., Henzler, M. & Heiland, G. Low temperature sublimation processes from clean cleaved polar surfaces of zinc oxide crystals during first heating. *Surf. Sci.* **41**, 403–411 (1974).

Comment 18: “I think that is important to link these results to previous ones where it has been exposed that surface disorder and vacancies are the key for stabilising the ZnO polar surfaces.”

Reply: Following the reviewer’s suggestion, we tried to link our results to the previous works reporting the importance of vacancy formation and resulting entropy contribution in stabilizing oxide polar surfaces with citing the relevant references [R5, R12]. The corresponding sentences in the introduction and the discussion are highlighted in yellow on pages 3 and 17, respectively.

Reference:

R5. Capdevila-Cortada, M. & López, N. Entropic contributions enhance polarity compensation for CeO₂ (100) surfaces. *Nat. Mater.* **16**, 328–334 (2017).

R12. Mora-Fonz, D. *et al.* Why Are Polar Surfaces of ZnO Stable? *Chem. Mater.* **29**, 5306–5320 (2017).

Comment 19: “It is not clear to me. Please, clarify how the surface models were built. Did you run calculations on clean stoichiometric ZnO polar surfaces? If so, those models do not represent the ZnO polar surfaces since there is a big dipole across the c axis and surface energies are large (unstable), compared to the other ZnO surfaces. The first step to model ionic polar surfaces is to diminish as much as possible the dipole. Considering the formal charges of the Zn and O species, there should be an ionic transfer of ~24% Zn²⁺ of the (0001) surface to the (000-1) surface to cancel out the dipole. This agrees with the best fit to X-ray diffraction data (10.1016/S0169-4332(00)00172-0), which was obtained with 25% Zn²⁺ vacancies at the Zn terminated surface.”

Reply: As we already replied to the previous comments (comment 14 and 16), we are well aware of the importance of the cancelation of the dipole of ZnO (0001) polar surface. We confirmed the stabilization of the (0001) polar surface by Zn vacancy formation experimentally (Figure 1) and explained how the (0001) surface was handled in the atomic simulations. The DFT and MD calculations in this study have been carried out using the atomic models with different extent of Zn vacancies depending on the information we want to know. For example, the desorption energy barriers for the surface Zn and O atoms were calculated by using the pristine surface models without Zn vacancies for each orientation (we compensated the electric field

by passivating the O dangling bond on the O-terminated surface on the other side using 0.5 fractionally charged pseudo-hydrogen). Except these calculations, Zn vacancies are introduced in all the other calculations, for example, the calculation of the diffusion barrier of Zn vacancies (Fig. 3e) and the MD simulation showing the quasi-liquid formation by surface disorder (Fig. 6). We described the initial state of each simulation cell in the Method section explicitly.

Comment 20: ““We generated three samples with different vacancy distribution for each Zn deficient case, except the pure ZnO, and the calculated elastic constants for the samples are averaged.” –what did you use the pure ZnO models for? What is the different vacancy distribution? Did you transfer those ionic charges to the O-terminated surface or just created the vacancies?”

Reply: First, we would like to clarify the purpose of the elastic constant calculation. Different from the stabilization mechanism of the (0001) polar surface by Zn vacancies, in this calculation we introduced Zn vacancies to the ZnO bulk supercell (no surface exists) to investigate how the lattice rigidity is affected by the Zn vacancies when they grow beyond the equilibrium surface concentration (25% Zn vacancy at Zn-terminated surface) and diffuse inward as we witnessed the large, anisotropic atomic displacement and subsequent disordering of ZnO lattice during the evaporation. Note that it is very difficult to calculate the elastic constants directly using the anisotropic slab cells with the gradient of the Zn vacancy concentration.

Even in the same Zn vacancy concentration, the elastic constants are slightly changed depending on the vacancy distribution in the supercell. For better statistics, we considered the three different supercells with randomly distributed V_{Zn} for each vacancy concentration, as one example is illustrated in Fig. R8, and the calculated elastic constants are averaged. The pure ZnO serves as the reference state for the elastic constants without Zn vacancies. We provided the details of the calculation in the Method section and included Fig. R8 in the supplementary information (Fig. S15).

Fig. R8. The three different supercells for 4% Zn vacancy concentration (two Zn vacancies in the cell) used in the elastic constant calculations. The positions of the Zn vacancies are denoted by blue circles, denoted by V1 and V2.

Comment 21: “What is the reason for using two different functionals (LDA+U and

PBE)? Why not to use either or to be more consistent? For example, on the latter, there is no account for the on-site Coulomb energy. I understand that CP2K could provide faster calculations for the AIMD simulations, then, one could use CP2K and PBE+U (or hybrids, not that expensive anymore on CP2K) and be more consistent with the results. After the AIMD simulation with CP2K+PBE the authors go back to VASP (with LDA+U or PBE? – it is not clear in the text). Calculations need to be consistent to provide an easier and fair comparison among them.”

Reply: We appreciate the reviewer for pointing out this. We admit that this was not clearly discussed in the original manuscript. It is well known that the LDA+U method with parameter $U = 4.7$ eV describes the defects in ZnO fairly well [R13], so we used the method to investigate the energetics of V_{Zn} . However, the composition and structure of the quasi-liquid layer are quite different from ZnO. As such, we do not know the empirical parameter U and the LDA+U method is not guaranteed for the quasi-liquid layer. In this regard, we used the non-empirical and unbiased DFT method. We have included the discussion in the Method section on page 20 to read “... This LDA+U method has been employed in previous works, and the results have shown that the method describes defects in ZnO fairly well. ...” and on page 22 to read “LDA+U method is well established for crystalline ZnO. Unlike the crystalline ZnO, empirical parameter U for LDA+U method is not known for metallic quasi-liquid ZnO ...”.

Reference:

R13. Janotti, A. & de Walle, C. G. Native point defects in ZnO. *Phys. Rev. B* **76**, 165202 (2007).

Reviewer #3

The manuscript deals with the study of ZnO materials and how they equilibrate when cut through a polar surface. The work is carefully done and presents important implications. However, I have severe reserves on the modeling part.

Reply: We thank the reviewer for his/her constructive and insightful comments that immensely helped us to improve the manuscript. Our detailed responses to the questions and comments are appended below:

Comment 1: “The use of LDA+U is really not justified, better functionals can be employed for this purpose and since the semiconductor nature of the native material is so relevant I think that either severe benchmarking or a better functional even including hybrid methods are needed to certify the relevance of the values obtained.”

Reply: It is well known that the LDA+U method describes energetics of defects in ZnO fairly well with reasonable computational cost [R13], so we used the method. We admit that this was not clearly discussed in the original manuscript, and we have included the discussion in the revised manuscript on page 20 to read “... This LDA+U method has been employed in previous works, and the results have shown that the method describes defects in ZnO fairly well. ...”

We have checked the energetics of desorption and V_{Zn} diffusion processes using HSE functional. Because our slab supercells (containing 188 atoms with 25 Å vacuum gap and requiring a 4×2×1 k-points mesh) is too large for hybrid functional calculations, we used the relaxed structures in the LDA+U functional calculations. As summarized in Table R1, the LDA+U results are qualitatively similar to the HSE results. We have included these results in the Supplementary information (Supplementary Note 6).

Table R1. Comparison of the LDA+U and HSE functional calculations.

	Desorption		Zn diffusion (layer)				
	Zn	O	0	1	2	3	4
LDA+U	0.3 eV	5.1 eV	0.0 eV	-0.2 eV	0.0 eV	0.4 eV	0.7 eV
HSE	0.4 eV	4.8 eV	0.0 eV	-0.3 eV	0.1 eV	0.5 eV	0.9 eV

Reference:

R13. Janotti, A. & de Walle, C. G. Native point defects in ZnO. *Phys. Rev. B* **76**, 165202 (2007).

Comment 2: “The nature of the diffuse layer formed is not seen in the computational part, can diffusion and aggregation be computed there?”

Reply: We rationalized the formation of quasi-liquid in the simulation by introducing

the Zn vacancies to a simulation slab and relaxing with a microscopic (NVE) ensemble. To identify the changes in crystal structure, the radial distribution functions were carried out from the structure obtained after 0.45 ps of MD simulation. The simulation details have been given in the Method section, “**Ab initio MD simulations of ZnO slab with quasi-liquid layer**” on page 21. The diffusion of Zn atoms in the quasi-liquid layer are assumed at the initial configuration by set up the Zn vacancies down to the top 3 Zn-O bilayers. The aggregation of V_{Zn} is unlikely due to their Coulombic repulsion [R14]. So, we consider the V_{Zn} with random distribution rather than the V_{Zn} clustering. To make the point clear, we have included the discussion on page 21, to read “**39 Zn atoms were randomly removed from the top 3 Zn-O bilayers...**”

Reference:

R14. Bang, J., Kim, Y.-S., Park, C. H., Gao, F. & Zhang, S. B. Understanding the presence of vacancy clusters in ZnO from a kinetic perspective. *Appl. Phys. Lett.* **104**, 252101 (2014).

Comment 3: Some of the energies are far to large to be representative. The computational part needs to indicate the reference energies employed.”

Reply: In the previous submission, we represented the desorption *barriers* for the desorption processes, so sometimes they are quite large. The reason that we used the desorption barrier is that we considered the kinetic process of the Zn or O desorption. Note that the same method had been used in a previous work [R1], where the kinetics of absorption and desorption of hydrogen on graphite had been considered. We admitted that some energies are too large to be representative, as the reviewer mentioned, and for better comparison to other works we have changed the values (desorption barriers) to “desorption energy”. We describe how a desorption energy is related to the corresponding desorption barrier below:

Because there is no uphill in the energy landscapes of the desorption processes [See Fig. S9], the desorption barrier of an atom X $E_{DB}(X)$ was calculated by $E_{DB}(X) = E(S:V_X + X^{away}) - E(S)$, where $E(S)$ and $E(S:V_X + X^{away})$ are the total energies of the pure surface and the surface containing a X vacancy (V_X) with a X atom far away from the surface, respectively. Here, $E(S:V_X + X^{away})$ is approximately the sum of the total energies of the surface with V_X $E(S:V_X)$ and the isolated X atom $E(X)$, i.e., $E(S:V_X + X^{away}) \cong E(S:V_X) + E(X)$, because the X atom is far away from the surface and considered as an isolated one. In our calculations, the energy differences between the left- and right-hand sides are below or about 0.1 eV for all the cases. On the other hand, the desorption energy of the X atom $E_{DE}(X)$ can be calculated by $E_{DE}(X) = E(S:V_X) + E_{ref}(X) - E(S)$, where $E_{ref}(X)$ is the reference energy of the X atom, i.e., $E_{ref}(Zn)$ and $E_{ref}(O)$ are the single atom energies of the Zn metal and O_2 molecule, respectively. Thus, the difference $E_{DB}(X) - E_{DE}(X) \cong E(X) - E_{ref}(X) = \Delta E(X)$, which is the energy change of a Zn or O atom associated with the formation of a Zn metal or O_2 molecule. In short, the desorption energies $E_{DE}(X)$ are smaller than the corresponding desorption barriers $E_{DB}(X)$ by the energy change $\Delta E(X)$, i.e., $E_{DE}(X) \cong E_{DB}(X) - \Delta E(X)$. In our calculations, $\Delta E(Zn)=1.8$ eV and $\Delta E(O)=3.3$ eV. We have included this discussion in the caption of Fig. S9.

Reference:

R1. Hornekaer, L. *et al.* Clustering of Chemisorbed H(D) Atoms on the Graphite (0001) Surface due to Preferential Sticking. *Phys. Rev. Lett.* **97**, 186102 (2006).

Comment 4: “4. The structures need to be uploaded to a repository for other scientist to reuse them.”

Reply: We have uploaded the atomic structures of the calculated surfaces in the VASP program format.

REVIEWERS' COMMENTS

Reviewer #2 (Remarks to the Author):

I am happy with the responses that the authors provided. I recommend the current version be published in Nature Communications.

Reviewer #3 (Remarks to the Author):

The authors have successfully addressed all the issues I raised in previous correspondence. Therefore, I can recommend the manuscript for publication.

Reviewer #4 (Remarks to the Author):

The authors have addressed all issues that were raised satisfactorily. We feel that the paper is now acceptable for publication in Nature Communications.

Minor points: Please let the paper be reread by a native speaker to fix many little mistakes such as articles missing. Two examples: The first sentence of the introduction has some problems: missing an article at the beginning + the verb is wrong + consisted-> consisting? Line 69: thereby decrease -> thereby decreasing?

Reviewer #2

I am happy with the responses that the authors provided. I recommend the current version be published in Nature Communications.

Reply: We thank the reviewer for positively evaluating the revised manuscript.

Reviewer #3

The authors have successfully addressed all the issues I raised in previous correspondence. Therefore, I can recommend the manuscript for publication.

Reply: We thank the reviewer for positively evaluating the revised manuscript.

Reviewer #4

The authors have addressed all issues that were raised satisfactorily. We feel that the paper is now acceptable for publication in Nature Communications.

Reply: We thank the reviewer for positively evaluating the revised manuscript.

Minor points: Please let the paper be reread by a native speaker to fix many little mistakes such as articles missing. Two examples: The first sentence of the introduction has some problems: missing an article at the beginning + the verb is wrong + consisted-> consisting? Line 69: thereby decrease -> thereby decreasing?

Reply: We have corrected what the reviewer pointed out and tried to fix mistakes throughout the manuscript as much as we can.